# Scaling the Convex Barrier with Active Sets

**Alessandro De Palma**[∗]**, Harkirat Singh Behl**[∗]**, Rudy Bunel, Philip H.S. Torr,**
**M. Pawan Kumar**
University of Oxford
{adepalma,harkirat,phst,pawan}@robots.ox.ac.uk
bunel.rudy@gmail.com

## Abstract

Tight and efficient neural network bounding is of critical importance for the scaling of neural network verification systems. A number of efficient specialised dual solvers for neural network bounds have been presented recently, but they are often too loose to verify more challenging properties. This lack of tightness is linked to the weakness of the employed relaxation, which is usually a linear program of size linear in the number of neurons. While a tighter linear relaxation for piecewise linear activations exists, it comes at the cost of exponentially many constraints and thus currently lacks an efficient customised solver. We alleviate this deficiency via a novel dual algorithm that realises the full potential of the new relaxation by operating on a small active set of dual variables. Our method recovers the strengths of the new relaxation in the dual space: tightness and a linear separation oracle. At the same time, it shares the benefits of previous dual approaches for weaker relaxations: massive parallelism, GPU implementation, low cost per iteration and valid bounds at any time. As a consequence, we obtain better bounds than off-the-shelf solvers in only a fraction of their running time and recover the speed-accuracy trade-offs of looser dual solvers if the computational budget is small. We demonstrate that this results in significant formal verification speed-ups.

## 1 Introduction

Verification requires formally proving or disproving that a given property of a neural network holds over all inputs in a specified domain. We consider properties in their canonical form (Bunel et al., 2018), which requires us to either: (i) prove that no input results in a negative output (property is true); or (ii) identify a counter-example (property is false). The search for counter-examples is typically performed by efficient methods such as random sampling of the input domain (Webb et al., 2019), or projected gradient descent (Carlini & Wagner, 2017). In contrast, establishing the veracity of a property requires solving a suitable convex relaxation to obtain a lower bound on the minimum output. If the lower bound is positive, the given property is true. If the bound is negative and no counter-example is found, either: (i) we make no conclusions regarding the property (incomplete verification); or (ii) we further refine the counter-example search and lower bound computation within a branch-and-bound framework until we reach a concrete conclusion (complete verification).

The main bottleneck of branch and bound is the computation of the lower bound for each node of the enumeration tree via convex optimization. While earlier works relied on off-the-shelf solvers (Ehlers, 2017; Bunel et al., 2018), it was quickly established that such an approach does not scale-up elegantly with the size of the neural network. This has motivated researchers to design specialized dual solvers (Dvijotham et al., 2019; Bunel et al., 2020a), thereby providing initial evidence that verification can be realised in practice. However, the convex relaxation considered in the dual solvers is itself very weak (Ehlers, 2017), hitting what is now commonly referred to as the "convex barrier" (Salman et al., 2019). In practice, this implies that either several properties remain undecided in incomplete verification, or take several hours to be verified exactly.

Multiple works have tried to overcome the convex barrier for piecewise linear activations (Raghunathan et al., 2018; Singh et al., 2019). Here, we focus on the single-neuron Linear Programming (LP) relaxation by Anderson et al. (2020). Unfortunately, its tightness comes at the price of exponentially many (in the number of variables) constraints. Therefore, existing dual solvers (Dvijotham et al., 2018; Bunel et al., 2020a) are not easily applicable, limiting the scaling of the new relaxation.

---

[∗]Equal contribution.

We address this problem by presenting a specialized dual solver for the relaxation by Anderson et al. (2020), which realises its full potential by meeting the following desiderata:

- By keeping an *active set* of dual variables, we obtain a *sparse* dual solver that recovers the strengths of the original primal problem (Anderson et al., 2020) in the dual domain. In line with previous dual solvers, our approach yields valid bounds at anytime, leverages convolutional network structure and enjoys *massive parallelism* within a GPU implementation, resulting in better bounds in an order of magnitude less time than off-the-shelf solvers (Gurobi Optimization, 2020).

- We present a unified dual treatment that includes both a linearly sized LP relaxation (Ehlers, 2017) and the tighter formulation. As a consequence, our solver provides a *wide range of speed-accuracy trade-offs*: (i) it is competitive with dual approaches on the looser relaxation (Dvijotham et al., 2018; Bunel et al., 2020a); and (ii) it yields much tighter bounds if a larger computational budget is available. Owing to this flexibility, we show that our dual algorithm yields large complete verification gains compared to primal approaches (Anderson et al., 2020) and previous dual algorithms.

## 2 PRELIMINARIES: NEURAL NETWORK RELAXATIONS

We denote vectors by bold lower case letters (for example, $\mathbf{x}$) and matrices by upper case letters (for example, $W$). We use $\odot$ for the Hadamard product, $[\![\cdot]\!]$ for integer ranges, $\mathbb{1}_{\mathbf{a}}$ for the indicator vector on condition $\mathbf{a}$ and brackets for intervals ($[\mathbf{l}_k, \mathbf{u}_k]$) and vector or matrix entries ($\mathbf{x}[i]$ or $W[i,j]$). In addition, given $W \in \mathbb{R}^{m \times n}$ and $\mathbf{x} \in \mathbb{R}^m$, we will employ $W \diamond \mathbf{x}$ and $W \square \mathbf{x}$ as shorthands for respectively $\sum_i \mathrm{col}_i(W) \odot \mathbf{x}$ and $\sum_i \mathrm{col}_i(W)^T \mathbf{x}$, where $\mathrm{col}_i(W)$ denotes the $i$-th column of matrix $W$.

Let $\mathcal{C}$ be the network input domain. Similar to Dvijotham et al. (2018); Bunel et al. (2020a), we assume that linear minimisation over $\mathcal{C}$ can be performed in closed-form. Our goal is to compute bounds on the scalar output of a piecewise-linear feedforward neural network. The tightest possible lower bound can be obtained by solving the following optimization problem:

$$\min_{\mathbf{x},\hat{\mathbf{x}}} \quad \hat{x}_n \qquad \text{s.t.} \quad \mathbf{x}_0 \in \mathcal{C}, \tag{1a}$$

$$\hat{\mathbf{x}}_{k+1} = W_{k+1}\mathbf{x}_k + \mathbf{b}_{k+1} \quad k \in [\![0, n-1]\!], \tag{1b}$$

$$\mathbf{x}_k = \sigma(\hat{\mathbf{x}}_k) \qquad k \in [\![1, n-1]\!], \tag{1c}$$

where the activation function $\sigma(\hat{\mathbf{x}}_k)$ is piecewise-linear, $\hat{\mathbf{x}}_k, \mathbf{x}_k \in \mathbb{R}^{n_k}$ denote the outputs of the $k$-th linear layer (fully-connected or convolutional) and activation function respectively, $W_k$ and $\mathbf{b}_k$ denote its weight matrix and bias, $n_k$ is the number of activations at layer k. We will focus on the ReLU case ($\sigma(\mathbf{x}) = \max(\mathbf{x}, 0)$), as common piecewise-linear functions can be expressed as a composition of ReLUs (Bunel et al., 2020b).

Problem (1) is non-convex due to the activation function's non-linearity (1c). As solving it is NP-hard (Katz et al., 2017), it is commonly approximated by a convex relaxation (see §4). The quality of the corresponding bounds, which is fundamental in verification, depends on the tightness of the relaxation. Unfortunately, tight relaxations usually correspond to slower bounding procedures. We first review a popular ReLU relaxation in §2.1). We then consider a tighter one in §2.2.

### 2.1 PLANET RELAXATION

The so-called Planet relaxation (Ehlers, 2017) has enjoyed widespread use due to its amenability to efficient customised solvers (Dvijotham et al., 2018; Bunel et al., 2020a) and is the "relaxation of choice" for many works in the area (Bunel et al., 2020b; Lu & Kumar, 2020). Here, we describe it in its non-projected form $\mathcal{M}_k$, the LP relaxation of the Big-M Mixed Integer Programming (MIP) formulation (Tjeng et al., 2019). Applying $\mathcal{M}_k$ to problem (1) results in:

$$\min_{\mathbf{x},\hat{\mathbf{x}},\mathbf{z}} \quad \hat{x}_n \quad \text{s.t.} \quad \mathbf{x}_0 \in \mathcal{C}$$

$$\left.\begin{array}{l} \hat{\mathbf{x}}_{k+1} = W_{k+1}\mathbf{x}_k + \mathbf{b}_{k+1} \\ \mathbf{x}_k \geq \hat{\mathbf{x}}_k, \quad \mathbf{x}_k \leq \hat{\mathbf{u}}_k \odot \mathbf{z}_k, \\ \mathbf{x}_k \leq \hat{\mathbf{x}}_k - \hat{\mathbf{l}}_k \odot (1 - \mathbf{z}_k), \\ (\mathbf{x}_k, \hat{\mathbf{x}}_k, \mathbf{z}_k) \in [\mathbf{l}_k, \mathbf{u}_k] \times [\hat{\mathbf{l}}_k, \hat{\mathbf{u}}_k] \times [\mathbf{0}, \mathbf{1}] \end{array}\right\} := \mathcal{M}_k \quad \begin{array}{l} k \in [\![0, n-1]\!], \\ \\ k \in [\![1, n-1]\!], \end{array} \tag{2}$$

where $\hat{\mathbf{l}}_k, \hat{\mathbf{u}}_k$ and $\mathbf{l}_k, \mathbf{u}_k$ are *intermediate bounds* respectively on pre-activation variables $\hat{\mathbf{x}}_k$ and post-activation variables $\mathbf{x}_k$. These constants play an important role in the structure of $\mathcal{M}_k$ and,

together with the relaxed binary constraints on $\mathbf{z}$, define box constraints on the variables. We detail how to compute intermediate bounds in appendix E. Projecting out auxiliary variables $\mathbf{z}$ results in the Planet relaxation (cf. appendix B.1 for details), which replaces (1c) by its convex hull.

Problem (2), which is linearly-sized, can be easily solved via commercial black-box LP solvers (Bunel et al., 2018). This does not scale-up well with the size of the neural network, motivating the need for specialised solvers. Customised dual solvers have been designed by relaxing constraints (1b), (1c) (Dvijotham et al., 2018) or replacing (1c) by the Planet relaxation and employing Lagrangian Decomposition (Bunel et al., 2020a). Both approaches result in bounds very close to optimality for problem (2) in only a fraction of the runtime of off-the-shelf solvers.

## 2.2 A Tighter Relaxation

A much tighter approximation of problem (1) than the Planet relaxation (§2.1) can be obtained by representing the convex hull of the composition of (1b) and (1c) rather than the convex hull of (1c) alone. A formulation of this type was recently introduced by Anderson et al. (2020).

Let us define $\check{L}_{k-1}, \check{U}_{k-1} \in \mathbb{R}^{n_k \times n_{k-1}}$ as: $\check{L}_{k-1}[i,j] = \mathbf{l}_{k-1}[j]\mathbb{1}_{W_k[i,j]\geq 0} + \mathbf{u}_{k-1}[j]\mathbb{1}_{W_k[i,j]<0}$, and $\check{U}_{k-1}[i,j] = \mathbf{u}_{k-1}[j]\mathbb{1}_{W_k[i,j]\geq 0} + \mathbf{l}_{k-1}[j]\mathbb{1}_{W_k[i,j]<0}$. Additionally, let us introduce $2^{W_k} = \{0,1\}^{n_k \times n_{k-1}}$, the set of all possible binary masks of weight matrix $W_k$, and $\mathcal{E}_k := 2^{W_k} \setminus \{0,1\}$, which excludes the all-zero and all-one masks. The new representation results in the following primal problem:

$$
\min_{\mathbf{x},\hat{\mathbf{x}},\mathbf{z}} \hat{x}_n \quad \text{s.t.} \quad \mathbf{x}_0 \in \mathcal{C}
$$

$$
\hat{\mathbf{x}}_{k+1} = W_{k+1}\mathbf{x}_k + \mathbf{b}_{k+1} \qquad\qquad k \in [\![0, n-1]\!],
$$

$$
\left.\begin{array}{l}
(\mathbf{x}_k, \hat{\mathbf{x}}_k, \mathbf{z}_k) \in \mathcal{M}_k \\[4pt]
\mathbf{x}_k \leq \begin{pmatrix} (W_k \odot I_k)\,\mathbf{x}_{k-1} + \mathbf{z}_k \odot \mathbf{b}_k \\ -\left(W_k \odot I_k \odot \check{L}_{k-1}\right) \diamond (1-\mathbf{z}_k) \\ +\left(W_k \odot (1-I_k) \odot \check{U}_{k-1}\right) \diamond \mathbf{z}_k \end{pmatrix} \quad \forall I_k \in \mathcal{E}_k
\end{array}\right\} := \mathcal{A}_k \quad k \in [\![1, n-1]\!].
$$

(3)

Both $\mathcal{M}_k$ and $\mathcal{A}_k$ yield valid MIP formulations for problem (1) when imposing integrality constraints on $\mathbf{z}$. However, the LP relaxation of $\mathcal{A}_k$ will yield tighter bounds. In the worst case, this tightness comes at the cost of exponentially many constraints: one for each $I_k \in \mathcal{E}_k$. On the other hand, given a set of primal assignments $(\mathbf{x}, \mathbf{z})$ that are not necessarily feasible for problem (3), one can efficiently compute the most violated constraint (if any) at that point. The mask associated to such constraint can be computed in linear-time (Anderson et al., 2020) as:

$$
I_k[i,j] = \mathbb{1}^T_{\left((1-\mathbf{z}_k[i])\odot \check{L}_{k-1}[i,j] + \mathbf{z}_k[i]\odot \check{U}_{k-1}[i,j] - \mathbf{x}_{k-1}[i]\right)W_k[i,j]\geq 0}.
$$

(4)

We point out that $\mathcal{A}_k$ slightly differs from the original formulation of Anderson et al. (2020), which does not explicitly include pre-activation bounds $\hat{\mathbf{l}}_k, \hat{\mathbf{u}}_k$ (which we treat via $\mathcal{M}_k$). While this was implicitly addressed in practical applications (Botoeva et al., 2020), not doing so has a strong negative effect on bound tightness, possibly to the point of yielding looser bounds than problem (2). In appendix F, we provide an example in which this is the case and extend the original derivation by Anderson et al. (2020) to recover $\mathcal{A}_k$ as in problem (3).

Owing to the exponential number of constraints, problem (3) cannot be solved as it is. As outlined by Anderson et al. (2020), the availability of a linear-time separation oracle (4) offers a natural primal *cutting plane* algorithm, which can then be implemented in off-the-shelf solvers: solve the Big-M LP (2), then iteratively add the most violated constraints from $\mathcal{A}_k$ at the optimal solution. When applied to the verification of small neural networks via off-the-shelf MIP solvers, this leads to substantial gains with respect to the looser Big-M relaxation (Anderson et al., 2020).

## 3 An Efficient Dual Solver for the Tighter Relaxation

Inspired by the success of dual approaches on looser relaxations (Bunel et al., 2020a; Dvijotham et al., 2019), we show that the formal verification gains by Anderson et al. (2020) (see §2.2) scale to larger networks if we solve the tighter relaxation in the dual space. Due to the particular structure of the relaxation, a customised solver for problem (3) needs to meet a number of requirements.

**Fact 1.** *In order to replicate the success of previous dual algorithms on looser relaxations, we need a solver for problem* (3) *with the following properties: (i)* sparsity: *a memory cost linear in the number of network activations in spite of exponentially many constraints, (ii)* tightness: *the bounds*

*should reflect the quality of those obtained in the primal space, (iii)* anytime*: low cost per iteration and valid bounds at each step.*

The anytime requirement motivates dual solutions: any dual assignment yields a valid bound due to weak duality. Unfortunately, as shown in appendix A, neither of the two dual derivations by Bunel et al. (2020a); Dvijotham et al. (2018) readily satisfy all desiderata at once. Therefore, we need a completely different approach. Let us introduce dual variables $\boldsymbol{\alpha}, \boldsymbol{\beta}$ and functions thereof:

$$
\begin{aligned}
\boldsymbol{f}_k(\boldsymbol{\alpha}, \boldsymbol{\beta}) =\ & \boldsymbol{\alpha}_k - W_{k+1}^T \boldsymbol{\alpha}_{k+1} - \textstyle\sum_{I_k} \boldsymbol{\beta}_{k,I_k} + \sum_{I_{k+1}} (W_{k+1} \odot I_{k+1})^T \boldsymbol{\beta}_{k+1,I_{k+1}}, \\
\boldsymbol{g}_k(\boldsymbol{\beta}) =\ & \left[ \begin{array}{l} \sum_{I_k \in \mathcal{E}_k} \left(W_k \odot (1 - I_k) \odot \check{U}_{k-1}\right) \diamond \boldsymbol{\beta}_{k,I_k} + \boldsymbol{\beta}_{k,0} \odot \hat{\mathbf{u}}_k + \boldsymbol{\beta}_{k,1} \odot \hat{\mathbf{l}}_k \\ + \sum_{I_k \in \mathcal{E}_k} \left(W_k \odot I_k \odot \check{L}_{k-1}\right) \diamond \boldsymbol{\beta}_{k,I_k} + \sum_{I_k \in \mathcal{E}_k} \boldsymbol{\beta}_{k,I_k} \odot \mathbf{b}_k, \end{array} \right.
\end{aligned} \tag{5}
$$

where $\sum_{I_k}$ is a shorthand for $\sum_{I_k \in 2^{W_k}}$. Starting from primal (3), we relax all constraints in $\mathcal{A}_k$ except box constraints (see §2.1). We obtain the following dual problem (derivation in appendix C), where functions $\boldsymbol{f}_k, \boldsymbol{g}_k$ appear in inner products with primal variables $\mathbf{x}_k, \mathbf{z}_k$:

$$
\max_{(\boldsymbol{\alpha}, \boldsymbol{\beta}) \geq 0} d(\boldsymbol{\alpha}, \boldsymbol{\beta}) \qquad \text{where:} \qquad d(\boldsymbol{\alpha}, \boldsymbol{\beta}) := \min_{\mathbf{x}, \mathbf{z}} \mathcal{L}(\mathbf{x}, \mathbf{z}, \boldsymbol{\alpha}, \boldsymbol{\beta}),
$$

$$
\mathcal{L}(\mathbf{x}, \mathbf{z}, \boldsymbol{\alpha}, \boldsymbol{\beta}) = \left[ \begin{array}{l} \sum_{k=1}^{n-1} \mathbf{b}_k^T \boldsymbol{\alpha}_k - \sum_{k=0}^{n-1} \boldsymbol{f}_k(\boldsymbol{\alpha}, \boldsymbol{\beta})^T \mathbf{x}_k - \sum_{k=1}^{n-1} \boldsymbol{g}_k(\boldsymbol{\beta})^T \mathbf{z}_k \\ + \sum_{k=1}^{n-1} \left( \sum_{I_k \in \mathcal{E}_k} (W_k \odot I_k \odot \check{L}_{k-1}) \square \boldsymbol{\beta}_{k,I_k} + \boldsymbol{\beta}_{k,1}^T(\hat{\mathbf{l}}_k - \mathbf{b}_k) \right) \end{array} \right. \tag{6}
$$

$$
\text{s.t.} \qquad \mathbf{x}_0 \in \mathcal{C}, \qquad (\mathbf{x}_k, \mathbf{z}_k) \in [\mathbf{l}_k, \mathbf{u}_k] \times [\mathbf{0},\ \mathbf{1}] \qquad k \in [\![1, n-1]\!].
$$

This is again a challenging problem: the exponentially many constraints in the primal (3) are now associated to an exponential number of variables. Nevertheless, we show that the requirements of Fact 1 can be met by operating on a restricted version of dual (6). To this end, we present Active Set, a specialised solver for the relaxation by Anderson et al. (2020) that is sparse, anytime and yields bounds reflecting the tightness of the new relaxation. Starting from the dual of problem (2), our solver iteratively adds variables to a small active set of dual variables $\boldsymbol{\beta}_{\mathcal{B}}$ and solves the resulting reduced version of problem (6). We first describe our solver on a fixed $\boldsymbol{\beta}_{\mathcal{B}}$ and then outline how to iteratively modify the active set (§3.2). Pseudo-code can be found in appendix D.

### 3.1 ACTIVE SET SOLVER

We want to solve a version of problem (6) for which the sums over the $I_k$ masks of each layer $k$ are restricted to $\mathcal{B}_k \subseteq \mathcal{E}_k{}^1$, with $\mathcal{B} = \cup_k \mathcal{B}_k$. By keeping $\mathcal{B} = \emptyset$, we recover a novel dual solver for the Big-M relaxation (2) (explicitly described in appendix B), which is employed as initialisation. Setting $\boldsymbol{\beta}_{k,I_k} = 0, \forall I_k \in \mathcal{E}_k \setminus \mathcal{B}_k$ in (5), (6) and removing these from the formulation, we obtain:

$$
\begin{aligned}
\boldsymbol{f}_{\mathcal{B},k}(\boldsymbol{\alpha}, \boldsymbol{\beta}_{\mathcal{B}}) &= \left[ \begin{array}{l} \boldsymbol{\alpha}_k - W_{k+1}^T \boldsymbol{\alpha}_{k+1} - \sum_{I_k \in \mathcal{B}_k \cup \{0,1\}} \boldsymbol{\beta}_{k,I_k}, \\ + \sum_{I_{k+1} \in \mathcal{B}_{k+1} \cup \{0,1\}} (W_{k+1} \odot I_{k+1})^T \boldsymbol{\beta}_{k+1,I_{k+1}} \end{array} \right. \\
\boldsymbol{g}_{\mathcal{B},k}(\boldsymbol{\beta}_{\mathcal{B}}) &= \left[ \begin{array}{l} \sum_{I_k \in \mathcal{B}_k} \left(W_k \odot (1 - I_k) \odot \check{U}_{k-1}\right) \diamond \boldsymbol{\beta}_{k,I_k} + \boldsymbol{\beta}_{k,0} \odot \hat{\mathbf{u}}_k + \boldsymbol{\beta}_{k,1} \odot \hat{\mathbf{l}}_k \\ + \sum_{I_k \in \mathcal{B}_k} \left(W_k \odot I_k \odot \check{L}_{k-1}\right) \diamond \boldsymbol{\beta}_{k,I_k} + \sum_{I_k \in \mathcal{B}_k} \boldsymbol{\beta}_{k,I_k} \odot \mathbf{b}_k, \end{array} \right.
\end{aligned} \tag{7}
$$

along with the reduced dual problem:

$$
\max_{(\boldsymbol{\alpha}, \boldsymbol{\beta}_{\mathcal{B}}) \geq 0} d_{\mathcal{B}}(\boldsymbol{\alpha}, \boldsymbol{\beta}_{\mathcal{B}}) \qquad \text{where:} \qquad d_{\mathcal{B}}(\boldsymbol{\alpha}, \boldsymbol{\beta}_{\mathcal{B}}) := \min_{\mathbf{x}, \mathbf{z}} \mathcal{L}_{\mathcal{B}}(\mathbf{x}, \mathbf{z}, \boldsymbol{\alpha}, \boldsymbol{\beta}_{\mathcal{B}}),
$$

$$
\mathcal{L}_{\mathcal{B}}(\mathbf{x}, \mathbf{z}, \boldsymbol{\alpha}, \boldsymbol{\beta}_{\mathcal{B}}) = \left[ \begin{array}{l} \sum_{k=1}^{n-1} \mathbf{b}_k^T \boldsymbol{\alpha}_k - \sum_{k=0}^{n-1} \boldsymbol{f}_{\mathcal{B},k}(\boldsymbol{\alpha}, \boldsymbol{\beta}_{\mathcal{B}})^T \mathbf{x}_k - \sum_{k=1}^{n-1} \boldsymbol{g}_{\mathcal{B},k}(\boldsymbol{\beta}_{\mathcal{B}})^T \mathbf{z}_k \\ + \sum_{k=1}^{n-1} \left( \sum_{I_k \in \mathcal{B}_k} (W_k \odot I_k \odot \check{L}_{k-1}) \square \boldsymbol{\beta}_{k,I_k} + \boldsymbol{\beta}_{k,1}^T(\hat{\mathbf{l}}_k - \mathbf{b}_k) \right) \end{array} \right.
$$

$$
\text{s.t.} \qquad \mathbf{x}_0 \in \mathcal{C}, \qquad (\mathbf{x}_k, \mathbf{z}_k) \in [\mathbf{l}_k, \mathbf{u}_k] \times [\mathbf{0},\ \mathbf{1}] \qquad k \in [\![1, n-1]\!]. \tag{8}
$$

We can maximize $d_{\mathcal{B}}(\boldsymbol{\alpha}, \boldsymbol{\beta}_{\mathcal{B}})$, which is concave and non-smooth, via projected supergradient ascent or variants thereof, such as Adam (Kingma & Ba, 2015). In order to obtain a valid supergradient, we need to perform the inner minimisation over the primals. Thanks to the structure of problem (8), the optimisation decomposes over the layers. For $k \in [\![1, n-1]\!]$, we can perform the minimisation in closed-form by driving the primals to their upper or lower bounds depending on the sign of their coefficients:

$$
\mathbf{x}_k^* = \mathbb{1}_{\boldsymbol{f}_{\mathcal{B},k}(\boldsymbol{\alpha}, \boldsymbol{\beta}_{\mathcal{B}}) \geq 0} \odot \hat{\mathbf{u}}_k + \mathbb{1}_{\boldsymbol{f}_{\mathcal{B},k}(\boldsymbol{\alpha}, \boldsymbol{\beta}_{\mathcal{B}}) < 0} \odot \hat{\mathbf{l}}_k, \qquad \mathbf{z}_k^* = \mathbb{1}_{\boldsymbol{g}_{\mathcal{B},k}(\boldsymbol{\beta}_{\mathcal{B}}) \geq 0} \odot \mathbf{1}. \tag{9}
$$

---

[1] As dual variables $\boldsymbol{\beta}_{k,I_k}$ are indexed by $I_k$, $\mathcal{B} = \cup_k \mathcal{B}_k$ implicitly defines an active set of variables $\boldsymbol{\beta}_{\mathcal{B}}$.

The subproblem corresponding to $\mathbf{x}_0$ is different, as it involves a linear minimization over $\mathbf{x}_0 \in \mathcal{C}$:

$$\mathbf{x}_0^* \in \operatorname{argmin}_{\mathbf{x}_0} \quad \boldsymbol{f}_{\mathcal{B},0}(\boldsymbol{\alpha}, \boldsymbol{\beta}_{\mathcal{B}})^T \mathbf{x}_0 \qquad \text{s.t.} \quad \mathbf{x}_0 \in \mathcal{C}. \tag{10}$$

We assumed in § 2 that (10) can be performed efficiently. We refer the reader to Bunel et al. (2020a) for descriptions of the minimisation when $\mathcal{C}$ is a $\ell_\infty$ or $\ell_2$ ball, as common for adversarial examples. Given $(\mathbf{x}^*, \mathbf{z}^*)$ as above, the supergradient of $d_{\mathcal{B}}(\boldsymbol{\alpha}, \boldsymbol{\beta}_{\mathcal{B}})$ is a subset of the one for $d(\boldsymbol{\alpha}, \boldsymbol{\beta})$, given by:

$$\nabla_{\boldsymbol{\alpha}_k} d(\boldsymbol{\alpha}, \boldsymbol{\beta}) = W_k \mathbf{x}_{k-1}^* + \mathbf{b}_k - \mathbf{x}_k^*, \qquad \nabla_{\boldsymbol{\beta}_{k,0}} d(\boldsymbol{\alpha}, \boldsymbol{\beta}) = \mathbf{x}_k^* - \mathbf{z}_k^* \odot \hat{\mathbf{u}}_k,$$

$$\nabla_{\boldsymbol{\beta}_{k,1}} d(\boldsymbol{\alpha}, \boldsymbol{\beta}) = \mathbf{x}_k^* - \left(W_k \mathbf{x}_{k-1}^* + \mathbf{b}_k\right) + (1 - \mathbf{z}_k^*) \odot \hat{\mathbf{l}}_k,$$

$$\nabla_{\boldsymbol{\beta}_{k,I_k}} d(\boldsymbol{\alpha}, \boldsymbol{\beta}) = \begin{pmatrix} \mathbf{x}_k^* - (W_k \odot I_k)\, \mathbf{x}_{k-1}^* + \left(W_k \odot I_k \odot \check{L}_{k-1}\right) \diamond (1 - \mathbf{z}_k^*) \\ -\mathbf{z}_k^* \odot \mathbf{b}_k + \left(W_k \odot (1 - I_k) \odot \check{U}_{k-1}\right) \diamond \mathbf{z}_k^* \end{pmatrix} \quad I_k \in \mathcal{B}_k, \tag{11}$$

for each $k \in [\![0, n-1]\!]$. At each iteration, after taking a step in the supergradient direction, the dual variables are projected to the non-negative orthant by clipping negative values.

## 3.2 EXTENDING THE ACTIVE SET

We initialise the dual (6) with a tight bound on the Big-M relaxation by solving for $d_\emptyset(\boldsymbol{\alpha}, \boldsymbol{\beta}_\emptyset)$ in (8). To satisfy the tightness requirement in Fact 1, we then need to include constraints (via their Lagrangian multipliers) from the exponential family of $\mathcal{A}_k$ into $\mathcal{B}_k$. Our goal is to tighten them as much as possible while keeping the active set small to save memory and compute.

The active set strategy is defined by a *selection criterion* for the $I_k^*$ to be added[2] to $\mathcal{B}_k$, and the *frequency* of addition. In practice, *we add the variables maximising the entries of supergradient $\nabla_{\boldsymbol{\beta}_{k,I_k}} d(\boldsymbol{\alpha}, \boldsymbol{\beta})$ after a fixed number of dual iterations.* We now provide motivation for both choices.

**Selection criterion** The selection criterion needs to be computationally efficient. Thus, we proceed greedily and focus only on the immediate effect at the current iteration. Let us map a restricted set of dual variables $\boldsymbol{\beta}_{\mathcal{B}}$ to a set of dual variables $\boldsymbol{\beta}$ for the full dual (6). We do so by setting variables not in the active set to 0: $\boldsymbol{\beta}_{\bar{\mathcal{B}}} = 0$, and $\boldsymbol{\beta} = \boldsymbol{\beta}_{\mathcal{B}} \cup \boldsymbol{\beta}_{\bar{\mathcal{B}}}$. Then, for each layer $k$, we add the set of variables $\boldsymbol{\beta}_{k,I_k^*}$ maximising the corresponding entries of the supergradient of the full dual problem (6): $\boldsymbol{\beta}_{k,I_k^*} \in \operatorname{argmax}_{\boldsymbol{\beta}_{k,I_k}} \{\nabla_{\boldsymbol{\beta}_{k,I_k}} d(\boldsymbol{\alpha}, \boldsymbol{\beta})^T \mathbf{1}\}$. Therefore, we use the subderivatives as a proxy for short-term improvement on the full dual objective $d(\boldsymbol{\alpha}, \boldsymbol{\beta})$. Under a primal interpretation, our selection criterion involves a call to the separation oracle (4) by Anderson et al. (2020).

**Proposition 1.** *$\boldsymbol{\beta}_{k,I_k^*}$ (as defined above) represents the Lagrangian multipliers associated to the most violated constraints from $\mathcal{A}_k$ at $(\mathbf{x}^*, \mathbf{z}^*) \in \operatorname{argmin}_{\mathbf{x}, \mathbf{z}} \mathcal{L}_{\mathcal{B}}(\mathbf{x}, \mathbf{z}, \boldsymbol{\alpha}, \boldsymbol{\beta}_{\mathcal{B}})$, the primal minimiser of the current restricted Lagrangian.*

*Proof.* See appendix D.1. □

**Frequency** Finally, we need to decide the frequency at which to add variables to the active set.

**Fact 2.** *Assume we obtained a dual solution $(\boldsymbol{\alpha}^\dagger, \boldsymbol{\beta}_{\mathcal{B}}^\dagger) \in \operatorname{argmax} d_{\mathcal{B}}(\boldsymbol{\alpha}, \boldsymbol{\beta}_{\mathcal{B}})$ using Active Set on the current $\mathcal{B}$. Then $(\mathbf{x}^*, \mathbf{z}^*) \in \operatorname{argmin}_{\mathbf{x}, \mathbf{z}} \mathcal{L}_{\mathcal{B}}(\mathbf{x}, \mathbf{z}, \boldsymbol{\alpha}^\dagger, \boldsymbol{\beta}_{\mathcal{B}}^\dagger)$ is not necessarily an optimal primal solution for the primal of the current restricted dual problem (Sherali & Choi, 1996).*

The primal of $d_{\mathcal{B}}(\boldsymbol{\alpha}, \boldsymbol{\beta}_{\mathcal{B}})$ (restricted primal) is the problem obtained by setting $\mathcal{E}_k \leftarrow \mathcal{B}_k$ in problem (3). While the primal cutting plane algorithm by Anderson et al. (2020) calls the separation oracle (4) at the optimal solution of the current restricted primal, Fact 2 shows that our selection criterion leads to a different behaviour even at dual optimality for $d_{\mathcal{B}}(\boldsymbol{\alpha}, \boldsymbol{\beta}_{\mathcal{B}})$. Therefore, as we have no theoretical incentive to reach (approximate) subproblem convergence, we add variables after a fixed tunable number of supergradient iterations. Furthermore, we can add more than one variable "at once" by running the oracle (4) repeatedly for a number of iterations.

We conclude this section by pointing out that, while recovering primal optima is possible in principle (Sherali & Choi, 1996), doing so would require dual convergence on each restricted dual problem (8). As the main advantage of dual approaches (Dvijotham et al., 2018; Bunel et al., 2020a) is their ability to quickly achieve tight bounds (rather than formal optimality), adapting the selection criterion to mirror the primal cutting plane algorithm would defeat the purpose of Active Set.

---

[2]adding a single $I_k^*$ mask to $\mathcal{B}_k$ extends $\boldsymbol{\beta}_{\mathcal{B}}$ by $n_k$ variables: one for each neuron at layer $k$.

### 3.3 Implementation Details, Technical Challenges

Analogously to previous dual algorithms (Dvijotham et al., 2018; Bunel et al., 2020a), our approach can leverage the massive parallelism offered by modern GPU architectures in three different ways. First, we execute in parallel the computations of lower and upper bounds relative to all the neurons of a given layer. Second, in complete verification, we can batch over the different Branch and Bound (BaB) subproblems. Third, as most of our solver relies on standard linear algebra operations employed during the forward and backward passes of neural networks, we can exploit the highly optimised implementations commonly found in modern deep learning frameworks.

An exception are what we call "masked" forward/backward passes: operations of the form $(W_k \odot I_k) \, \mathbf{x}_k$ or $(W_k \odot I_k)^T \, \mathbf{x}_{k+1}$, which are needed whenever dealing with constraints from $\mathcal{A}_k$. In our solver, they appear if $\mathcal{B}_k \neq \emptyset$ (see equations (8), (11)). Masked passes require a customised lower-level implementation for a proper treatment of convolutional layers, detailed in appendix G.

## 4 Related Work

In addition to those described in §2, many other relaxations have been proposed in the literature. In fact, all bounding methods are equivalent to solving some convex relaxation of a neural network. This holds for conceptually different ideas such as bound propagation (Gowal et al., 2018), specific dual assignments (Wong & Kolter, 2018), dual formulations based on Lagrangian Relaxation (Dvijotham et al., 2018) or Decomposition (Bunel et al., 2020a). The degree of tightness varies greatly: from looser relaxations associated to closed-form methods (Gowal et al., 2018; Weng et al., 2018; Wong & Kolter, 2018) to tighter formulations based on Semi-Definite Programming (SDP) (Raghunathan et al., 2018).

The speed of closed-form approaches results from simplifying the triangle-shaped feasible region of the Planet relaxation (§2.1) (Singh et al., 2018; Wang et al., 2018). On the other hand, tighter relaxations are more expressive than the linearly-sized LP by (Ehlers, 2017). The SDP formulation by Raghunathan et al. (2018) can represent interactions between activations in the same layer. Similarly, Singh et al. (2019) tighten the Planet relaxation by considering the convex hull of the union of polyhedra relative to $k$ ReLUs of a given layer at once. Alternatively, tighter LPs can be obtained by considering the ReLU together with the affine operator before it: standard MIP techniques (Jeroslow, 1987) lead to a formulation that is quadratic in the number of variables (see appendix F.2). The relaxation by Anderson et al. (2020) detailed in §2.2 is a more convenient representation of the same set.

By projecting out the auxiliary $\mathbf{z}$ variables, (Tjandraatmadja et al., 2020) recently introduced another formulation equivalent to the one by Anderson et al. (2020), with half as many variables and a linear factor more constraints compared to what described in §2.2. Therefore, the relationship between the two formulations mirrors the one between the Planet and Big-M relaxations (see appendix B.1). Our dual derivation and the Active Set algorithm can be adapted to operate on the projected relaxations.

Specialised dual solvers significantly improve in bounding efficiency with respect to off-the-shelf solvers for both LP (Bunel et al., 2020a) and SDP formulations (Dvijotham et al., 2019). Therefore, the design of similar solvers for other tight relaxations is an interesting line of future research. We contribute with a specialised dual solver for the relaxation by Anderson et al. (2020) (§3). In what follows, we demonstrate empirically that by seamlessly transitioning from the Planet relaxation to the tighter formulation, we can obtain large incomplete and complete verification improvements.

## 5 Experiments

We empirically demonstrate the effectiveness of our method under two settings. On incomplete verification (§5.1), we assess the speed and quality of bounds compared to other bounding algorithms. On complete verification (§5.2), we examine whether our speed-accuracy trade-offs correspond to faster exact verification. Our implementation is based on Pytorch (Paszke et al., 2017) and is available at `https://github.com/oval-group/scaling-the-convex-barrier`.

### 5.1 Incomplete Verification

We evaluate incomplete verification performance by upper bounding the robustness margin (the difference between the ground truth logit and the other logits) to adversarial perturbations (Szegedy et al., 2014) on the CIFAR-10 test set (Krizhevsky & Hinton, 2009). If the upper bound is negative, we can certify the network's vulnerability to adversarial perturbations. We replicate the experimen-

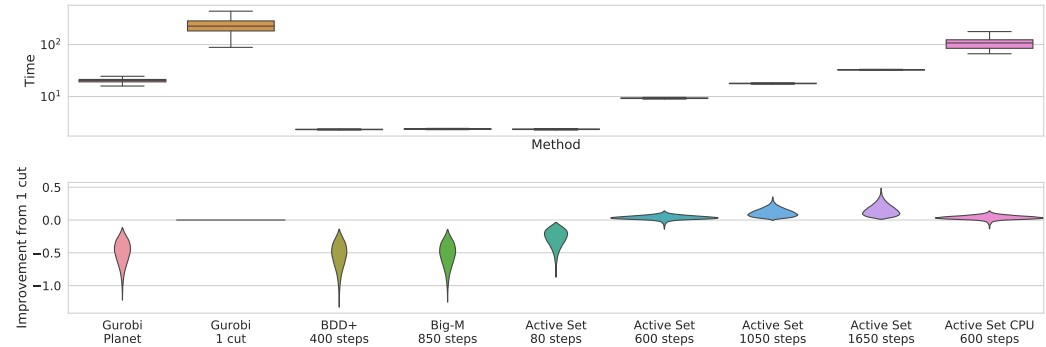

Figure 1: Upper plot: distribution of runtime in seconds. Lower plot: difference with the bounds obtained by Gurobi with a cut from $\mathcal{A}_k$ per neuron; higher is better. Results for the SGD-trained network from Bunel et al. (2020a). The width at a given value represents the proportion of problems for which this is the result. Comparing Active Sets with 1650 steps to Gurobi 1 Cut, tighter bounds are achieved with a smaller runtime.

tal setting from Bunel et al. (2020a). The networks correspond to the small network architecture from Wong & Kolter (2018). Here, we present results for a network trained via standard SGD and cross entropy loss, with no modification to the objective for robustness. Perturbations for this network lie in a $\ell_\infty$ norm ball with radius $\epsilon_{ver} = 5/255$ (which is hence lower than commonly employed radii for robustly trained networks). In appendix I, we provide additional CIFAR-10 results on an adversarially trained network using the method by Madry et al. (2018), and on MNIST (LeCun et al., 1998), for a network adversarially trained with the algorithm by Wong & Kolter (2018).

We compare both against previous dual iterative methods and Gurobi (Gurobi Optimization, 2020), the commercial black-box solver employed by Anderson et al. (2020). For Gurobi-based baselines, **Planet** means solving the Planet Ehlers (2017) relaxation of the network, while **Gurobi cut** starts from the Big-M relaxation and adds constraints from $\mathcal{A}_k$ in a cutting-plane fashion, as the original primal algorithm by Anderson et al. (2020). We run both on 4 CPU threads. Amongst dual iterative methods, run on an Nvidia Titan Xp GPU, we compare with **BDD+**, the recent proximal-based solver by Bunel et al. (2020a), operating on a Lagrangian Decomposition dual of the Planet relaxation. As we operate on (a subset of) the data by Bunel et al. (2020a), we omit both their supergradient-based approach and the one by Dvijotham et al. (2018), as they both perform worse than BDD+ (Bunel et al., 2020a). For the same reason, we omit cheaper (and looser) methods, like interval propagation Gowal et al. (2018) and the one by Wong & Kolter (2018). **Active Set** denotes our solver for problem (3), described in §3.1. By keeping $\mathcal{B} = \emptyset$, Active Set reduces to **Big-M**, a solver for the non-projected Planet relaxation (appendix B), which can be seen as Active Set's initialiser. In line with previous bounding algorithms (Bunel et al., 2020a), we employ Adam updates (Kingma & Ba, 2015) for supergradient-type methods due to their faster empirical convergence. Finally, we complement the comparison with Gurobi-based methods by running Active Set on 4 CPU threads (**Active Set CPU**). Further details, including hyper-parameters, can be found in appendix I.

Figure 1 shows the distribution of runtime and the bound improvement with respect to Gurobi cut for the SGD-trained network. For Gurobi cut, we only add the single most violated cut from $\mathcal{A}_k$ per neuron, due to the cost of repeatedly solving the LP. We tuned BDD+ and Big-M, the dual methods operating on the weaker relaxation (2), to have the same average runtime. They obtain bounds comparable to Gurobi Planet in one order less time. Initialised from 500 Big-M iterations, at 600

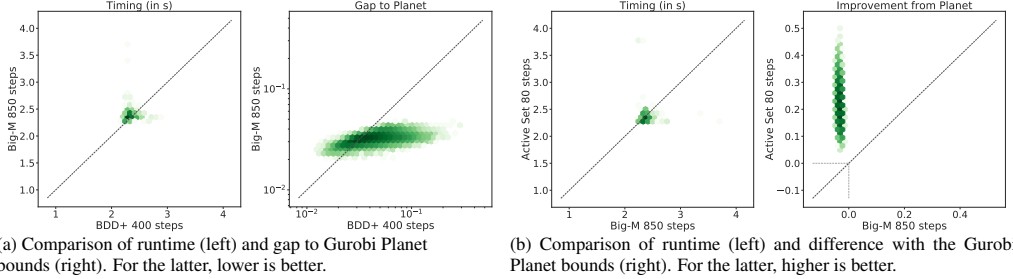

(a) Comparison of runtime (left) and gap to Gurobi Planet bounds (right). For the latter, lower is better.

(b) Comparison of runtime (left) and difference with the Gurobi Planet bounds (right). For the latter, higher is better.

Figure 2: Pointwise comparison for a subset of the methods on the data presented in Figure 1. Darker colour shades mean higher point density (on a logarithmic scale). The oblique dotted line corresponds to the equality.

iterations Active Set already achieves better bounds on average than Gurobi cut in around $1/20^{th}$ of the time. With a computational budget twice as large (1050 iterations) or four times as large (1650 iterations), the bounds significantly improve over Gurobi cut in still a fraction of the time. As we empirically demonstrate in appendix I, the tightness of the Active Set bounds is strongly linked to our active set strategy (§3.2). Remarkably, even if our method is specifically designed to take advantage of GPU acceleration, executing it on CPU proves to be strongly competitive with Gurobi cut, producing better bounds in less time for the benchmark of Figure 1.

Figure 2 shows pointwise comparisons for a subset of the methods of Figure 1, on the same data. Figure 2a shows the gap to the (Gurobi) Planet bound for BDD+ and our Big-M solver. Surprisingly, our Big-M solver is competitive with BDD+, achieving on average better bounds than BDD+, in the same time. Figure 2b shows the improvement over Planet bounds for Big-M and Active Set. The latter achieves markedly better bounds than Big-M in the same time, demonstrating the benefit of operating (at least partly) on the tighter dual (6).

## 5.2 COMPLETE VERIFICATION

We next evaluate the performance on complete verification, verifying the adversarial robustness of a network to perturbations in $\ell_\infty$ norm on a subset of the dataset by Lu & Kumar (2020), replicating the experimental setting from Bunel et al. (2020a). The dataset associates a different perturbation radius $\epsilon_{verif}$ to each CIFAR-10 image, so as to create challenging verification properties. Its difficulty makes the dataset an appropriate testing ground for tighter relaxations like the one by Anderson et al. (2020) (§2.2). Further details, including network architectures, can be found in appendix I.

Here, we aim to solve the non-convex problem (1) directly, rather than an approximation like §5.1. In order to do so, we use BaSBR, a branch and bound algorithm from Bunel et al. (2020b). Branch and Bound works by dividing the problem domain into subproblems (branching) and bounding the local minimum over those domains. Any domain which cannot contain the global lower bound is pruned away, whereas the others are kept and branched over. In BaBSR, branching is carried out by splitting an unfixed ReLU into its passing and blocking phases. The ReLU which induces maximum change in the domain's lower bound, when made unambiguous, is selected for splitting.

A fundamental component of a BaB method is the bounding algorithm, which is, in general, the computational bottleneck (Lu & Kumar, 2020). Therefore, we compare the effect on final verification time of using the different bounding methods in §5.1 within BaBSR. In addition, we evaluate **MIP** $\mathcal{A}_k$, which encodes problem (1) as a Big-M MIP (Tjeng et al., 2019) and solves it in Gurobi by adding cutting planes from $\mathcal{A}_k$, analogously to the original experiments by Anderson et al. (2020). Finally, we also compare against **ERAN** (Singh et al., 2020), a state-of-the-art complete verification toolbox: results on the dataset by Lu & Kumar (2020) are taken from the recent VNN-COMP competition (VNN-COMP, 2020). We use 100 iterations for Active Set, 100 iterations for BDD+ and 180 iterations for Big-M. For dual iterative algorithms, we solve 300 subproblems at once for the base network and 200 for the deep and wide networks (see §3.3). Additionally, dual variables are initialised from their parent node's bounding computation. As in Bunel et al. (2020a), the time-limit is kept at one hour. Due to the difference in computational cost between algorithms operating on the tighter relaxation by Anderson et al. (2020) and the other bounding algorithms[3], we also experiment with a *stratified* version of the bounding within BaBSR. We devise a set of heuristics to determine

---

[3]For Active Set, this is partly due to the masked forward/backward pass described in appendix G.

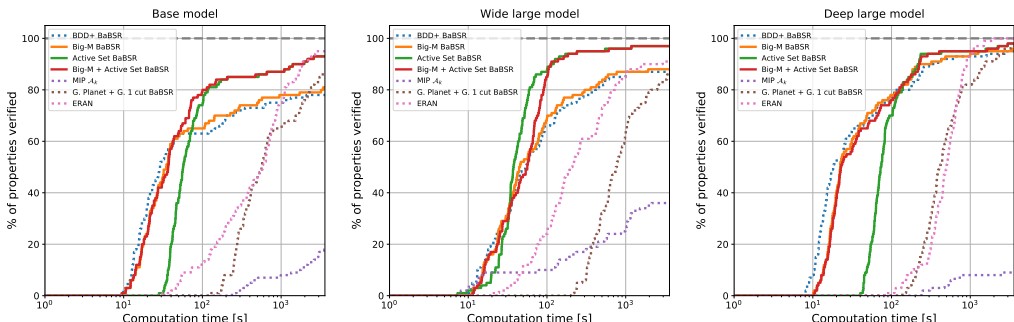

Figure 3: Cactus plots on properties from Lu & Kumar (2020), displaying the percentage of solved properties as a function of runtime. Baselines are represented by dotted lines.

| Method | Base | | | Wide | | | Deep | | |
|---|---|---|---|---|---|---|---|---|---|
| | time(s) | sub-problems | %Timeout | time(s) | sub-problems | %Timeout | time(s) | sub-problems | %Timeout |
| BDD+ BABSR | 883.55 | 82 699.40 | 22.00 | 568.25 | 43 751.88 | 13.00 | 281.47 | 10 763.48 | 5.00 |
| BIG-M BABSR | 826.60 | 68 582.00 | 19.00 | 533.79 | 35 877.24 | 12.00 | 253.37 | 9346.78 | 4.00 |
| A. SET BABSR | 422.32 | **9471.90** | 7.00 | **169.73** | **1873.36** | **3.00** | 227.26 | **2302.16** | **2.00** |
| BIG-M + A. SET BABSR | **402.88** | 11 408.84 | **7.00** | 179.73 | 3712.62 | **3.00** | **197.99** | 3086.62 | **2.00** |
| G. PLANET + G. 1 CUT BABSR | 1191.48 | 2044.28 | 14.00 | 1272.99 | 1352.42 | 10.00 | 704.59 | 677.74 | 3.00 |
| MIP $\mathcal{A}_k$ | 3227.50 | 226.24 | 82.00 | 2500.70 | 100.93 | 64.00 | 3339.37 | 434.57 | 91.00 |
| ERAN | 805.89 | - | 5.00 | 632.12 | - | 9.00 | 545.72 | - | 0.00 |

Table 1: We compare average solving time, average number of solved sub-problems and the percentage of timed out properties on data from Lu & Kumar (2020). The best dual iterative method is highlighted in bold.

whether a given subproblem is easy (therefore looser bounds are sufficient) or whether we need to operate on the tighter relaxation. Instances of this approach are **Big-M + Active Set** and **Gurobi Planet + Gurobi 1 cut**. Further details are provided in appendix H.

Figure 3 and Table 1 show that Big-M performs competitively with BDD+. Active Set verifies a larger share of properties than the methods operating on the looser formulation (2), demonstrating the benefit of tighter bounds (§5.1) in complete verification. On the other hand, the poor performance of MIP + $\mathcal{A}_k$ and of Gurobi Planet + Gurobi 1 cut, tied to scaling limitations of off-the-shelf solvers, shows that tighter bounds are effective only if they can be computed efficiently. Nevertheless, the difference in performance between the two Gurobi-based methods confirms that customised Branch and Bound solvers (BaBSR) are preferable to generic MIP solvers, as observed by Bunel et al. (2020b) on the looser Planet relaxation. Moreover, the stratified bounding system allows us to retain the speed of Big-M on easier properties, without excessively sacrificing Active Set's gains on the harder ones. Finally, while ERAN verifies 2% more properties than Active Set on two networks, BaBSR (with any dual bounding algorithm) is faster on most of the properties. BaBSR-based results could be further improved by employing the learned branching strategy presented by Lu & Kumar (2020): in this work, we focused on the bounding component of branch and bound.

## 6 DISCUSSION

The vast majority of neural network bounding algorithms focuses on (solving or loosening) a popular triangle-shaped relaxation, referred to as the "convex barrier" for verification. Relaxations that are tighter than this convex barrier have been recently introduced, but their complexity hinders applicability. We have presented Active Set, a sparse dual solver for one such relaxation, and empirically demonstrated that it yields significant formal verification speed-ups. Our results show that scalable tightness is key to the efficiency of neural network verification and instrumental in the definition of a more appropriate "convex barrier". We believe that new customised solvers for similarly tight relaxations are a crucial avenue for future research in the area, possibly beyond piecewise-linear networks. Finally, as it is inevitable that tighter bounds will come at a larger computational cost, future verification systems will be required to recognise a priori whether tight bounds are needed for a given property. A possible solution to this problem could rely on learning algorithms.

ACKNOWLEDGMENTS

ADP was supported by the EPSRC Centre for Doctoral Training in Autonomous Intelligent Machines and Systems, grant EP/L015987/1, and an IBM PhD fellowship. HSB was supported using a Tencent studentship through the University of Oxford.

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

## A  LIMITATIONS OF PREVIOUS DUAL APPROACHES

In this section, we show that previous dual derivations (Bunel et al., 2020a; Dvijotham et al., 2018) violate Fact 1. Therefore, they are not efficiently applicable to problem (3), motivating our own derivation in section 3.

We start from the approach by Dvijotham et al. (2018), which relies on relaxing equality constraints (1b), (1c) from the original non-convex problem (1). Dvijotham et al. (2018) prove that this relaxation corresponds to solving convex problem (2), which is equivalent to the Planet relaxation (Ehlers, 2017), to which the original proof refers. As we would like to solve tighter problem (3), the derivation is not directly applicable. Relying on intuition from convex analysis applied to duality gaps (Lemaréchal, 2001), we conjecture that relaxing the composition (1c) ∘ (1b) might tighten the primal problem equivalent to the relaxation, obtaining the following dual:

$$
\max_{\boldsymbol{\mu}} \min_{\mathbf{x}} \quad W_n \mathbf{x}_{n-1} + \mathbf{b}_n + \sum_{k=1}^{n-1} \boldsymbol{\mu}_k^T \left( \mathbf{x}_k - \max\{W_k \mathbf{x}_{k-1} + \mathbf{b}_k, 0\} \right)
$$
$$
\text{s.t.} \quad \mathbf{l}_k \le \mathbf{x}_k \le \mathbf{u}_k \qquad k \in [\![1, n-1]\!], \tag{12}
$$
$$
\mathbf{x}_0 \in \mathcal{C}.
$$

Unfortunately dual (12) requires an LP (the inner minimisation over $\mathbf{x}$, which in this case does not decompose over layers) to be solved exactly to obtain a supergradient and any time a valid bound is needed. This is markedly different from the original dual by Dvijotham et al. (2018), which had an efficient closed-form for the inner problems.

The derivation by Bunel et al. (2020a), instead, operates by substituting (1c) with its convex hull and solving its Lagrangian Decomposition dual. The Decomposition dual for the convex hull of (1c) ∘ (1b) (i.e., $\mathcal{A}_k$) takes the following form:

$$
\max_{\boldsymbol{\rho}} \min_{\mathbf{x}, \mathbf{z}} W_n \mathbf{x}_{A,n-1} + \mathbf{b}_n + \sum_{k=1}^{n-1} \boldsymbol{\rho}_k^T \left( \mathbf{x}_{B,k} - \mathbf{x}_{A,k} \right)
$$
$$
\text{s.t.} \quad \mathbf{x}_0 \in \mathcal{C}, \tag{13}
$$
$$
(\mathbf{x}_{B,k}, W_n \mathbf{x}_{A,n-1} + \mathbf{b}_n, \mathbf{z}_k) \in \mathcal{A}_{\text{dec},k} \qquad k \in [\![1, n-1]\!],
$$

where $\mathcal{A}_{\text{dec},k}$ corresponds to $\mathcal{A}_k$ with the following substitutions: $\mathbf{x}_k \rightarrow \mathbf{x}_{B,k}$, and $\hat{\mathbf{x}}_k \rightarrow W_n \mathbf{x}_{A,n-1} + \mathbf{b}_n$. It can be easily seen that the inner problems (the inner minimisation over $\mathbf{x}_{A,k}, \mathbf{x}_{B,k}$, for each layer $k > 0$) are an exponentially sized LP. Again, this differs from the original dual on the Planet relaxation (Bunel et al., 2020a), which had an efficient closed-form for the inner problems.

## B  DUAL INITIALISATION

---

**Algorithm 1** Big-M solver

---

1: **function** BIGM_COMPUTE_BOUNDS($\{W_k, \mathbf{b}_k, \mathbf{l}_k, \mathbf{u}_k, \hat{\mathbf{l}}_k, \hat{\mathbf{u}}_k\}_{k=1..n}$)
2:    Initialise duals $\boldsymbol{\alpha}^0, \boldsymbol{\beta}_{\mathcal{M}}^0$ using interval propagation bounds (Gowal et al., 2018)
3:    **for** $t \in [\![1, T-1]\!]$ **do**
4:        $\mathbf{x}^*, \mathbf{z}^* \in \operatorname{argmin}_{\mathbf{x}, \mathbf{z}} \mathcal{L}_{\mathcal{M}}(\mathbf{x}, \mathbf{z}, \boldsymbol{\alpha}^t, \boldsymbol{\beta}_{\mathcal{M}}^t)$ using (16)-(17)
5:        Compute supergradient using (18)
6:        $\boldsymbol{\alpha}^{t+1}, \boldsymbol{\beta}_{\mathcal{M}}^{t+1} \leftarrow$ Adam's update rule (Kingma & Ba, 2015)
7:        $\boldsymbol{\alpha}^{t+1}, \boldsymbol{\beta}_{\mathcal{M}}^{t+1} \leftarrow \max(\boldsymbol{\alpha}^{t+1}, 0), \max(\boldsymbol{\beta}_{\mathcal{M}}^{t+1}, 0)$        (dual projection)
8:    **end for**
9:    **return** $\min_{\mathbf{x}, \mathbf{z}} \mathcal{L}_{\mathcal{M}}(\mathbf{x}, \mathbf{z}, \boldsymbol{\alpha}^T, \boldsymbol{\beta}_{\mathcal{M}}^T)$
10: **end function**

---

As shown in section 3, our Active Set solver yields a dual solver for the Big-M relaxation (2) if the active set $\mathcal{B}$ is kept empty throughout execution. As indeed $\mathcal{B} = \emptyset$ for the first Active Set iterations (see algorithm 2 in section D), the Big-M solver can be thought of as dual initialisation. Furthermore,

we demonstrate experimentally in §5 that, when used as a stand-alone solver, our Big-M solver is competitive with previous dual algorithms for problem (2). The goal of this section is to explicitly describe the Big-M solver, which is summarised in algorithm 1. We point out that, in the notation of restricted variable sets from section 3.1, $\boldsymbol{\beta}_{\mathcal{M}} := \boldsymbol{\beta}_{\emptyset}$.

We now describe the equivalence between the Big-M and Planet relaxations, before presenting the solver in section B.3 and the dual it operates on in section B.2.

### B.1 EQUIVALENCE TO PLANET

As previously shown (Bunel et al., 2018), the Big-M relaxation ($\mathcal{M}_k$, when considering the $k$-th layer only) in problem (2) is equivalent to the Planet relaxation by Ehlers (2017). Then, due to strong duality, our Big-M solver (section B.2) and the solvers by Bunel et al. (2020a); Dvijotham et al. (2018) will all converge to the bounds from the solution of problem (2). In fact, the Decomposition-based method (Bunel et al., 2020a) directly operates on the Planet relaxation, while Dvijotham et al. (2018) prove that their dual is equivalent to doing so.

On the $k$-th layer, the Planet relaxation takes the following form:

$$
\mathcal{P}_k := \begin{cases}
\text{if } \hat{\mathbf{l}}_k \leq 0 \text{ and } \hat{\mathbf{u}}_k \geq 0 : \\
\qquad\qquad \mathbf{x}_k \geq 0, \quad \mathbf{x}_k \geq \hat{\mathbf{x}}_k, \\
\qquad\qquad \mathbf{x}_k \leq \hat{\mathbf{u}}_\mathbf{k} \odot (\hat{\mathbf{x}}_k - \hat{\mathbf{l}}_k) \odot \left(1/(\hat{\mathbf{u}}_\mathbf{k} - \hat{\mathbf{l}}_\mathbf{k})\right). \\
\text{if } \hat{\mathbf{u}}_k \leq 0 : \\
\qquad\qquad \mathbf{x}_k = 0. \\
\text{if } \hat{\mathbf{l}}_k \geq 0 : \\
\qquad\qquad \mathbf{x}_k = \hat{\mathbf{x}}_k.
\end{cases}
\tag{14}
$$

It can be seen that $\mathcal{P}_k = \text{Proj}_{\mathbf{x},\hat{\mathbf{x}}}(\mathcal{M}_k)$, where $\text{Proj}_{\mathbf{x},\hat{\mathbf{x}}}$ denotes projection on the $\mathbf{x}, \hat{\mathbf{x}}$ hyperplane. In fact, as $\mathbf{z}_k$ does not appear in the objective of the primal formulation (2), but only in the constraints, this means assigning it the value that allows the largest possible feasible region. This is trivial for passing or blocking ReLUs. For the ambiguous case, instead, Figure 4 (on a single ReLU) shows that $z_k = \frac{\hat{x}_k - \hat{l}_k}{\hat{u}_k - \hat{l}_k}$ is the correct assignment.

### B.2 BIG-M DUAL

As evident from problem (3), $\mathcal{A}_k \subseteq \mathcal{M}_k$. If we relax all constraints in $\mathcal{M}_k$ (except, again, the box constraints), we are going to obtain a dual with a strict subset of the variables in problem (6). The Big-M dual is a specific instance of the Active Set dual (8) where $\mathcal{B} = \emptyset$, and it takes the following

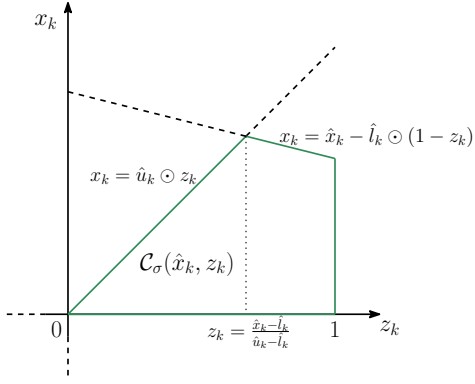

Figure 4: $\mathcal{M}_k$ plotted on the $(\mathbf{z}_k, \mathbf{x}_k)$ plane, under the assumption that $\hat{\mathbf{l}}_k \leq 0$ and $\hat{\mathbf{u}}_k \geq 0$.

form:

$$\max_{(\boldsymbol{\alpha},\boldsymbol{\beta})\geq 0} d_{\mathcal{M}}(\boldsymbol{\alpha},\boldsymbol{\beta}_{\mathcal{M}}) \qquad \text{where:} \qquad d_{\mathcal{M}}(\boldsymbol{\alpha},\boldsymbol{\beta}_{\mathcal{M}}) := \min_{\mathbf{x},\mathbf{z}} \ \mathcal{L}_{\mathcal{M}}(\mathbf{x},\mathbf{z},\boldsymbol{\alpha},\boldsymbol{\beta}_{\mathcal{M}}),$$

$$\mathcal{L}_{\mathcal{M}}(\mathbf{x},\mathbf{z},\boldsymbol{\alpha},\boldsymbol{\beta}_{\mathcal{M}}) = \begin{bmatrix} -\sum_{k=0}^{n-1}\left(\boldsymbol{\alpha}_k - W_{k+1}^T\boldsymbol{\alpha}_{k+1} - (\boldsymbol{\beta}_{k,0}+\boldsymbol{\beta}_{k,1}-W_{k+1}^T\boldsymbol{\beta}_{k+1,1})\right)^T\mathbf{x}_k \\ +\sum_{k=1}^{n-1}\mathbf{b}_k^T\boldsymbol{\alpha}_k - \sum_{k=1}^{n-1}\left(\boldsymbol{\beta}_{k,0}\odot\hat{\mathbf{u}}_k + \boldsymbol{\beta}_{k,1}\odot\hat{\mathbf{l}}_k\right)^T\mathbf{z}_k \\ +\sum_{k=1}^{n-1}(\hat{\mathbf{l}}_k-\mathbf{b}_k)^T\boldsymbol{\beta}_{k,1} \end{bmatrix}$$

$$\text{s.t.} \qquad \mathbf{x}_0 \in \mathcal{C}, \qquad (\mathbf{x}_k,\mathbf{z}_k) \in [\mathbf{l}_k,\mathbf{u}_k]\times[\mathbf{0},\ \mathbf{1}] \qquad k\in[\![1,n-1]\!]. \tag{15}$$

## B.3 BIG-M SOLVER

We initialise dual variables from interval propagation bounds (Gowal et al., 2018): this can be easily done by setting all dual variables except $\boldsymbol{\alpha}_n$ to 0. Then, we can maximize $d_{\mathcal{M}}(\boldsymbol{\alpha},\boldsymbol{\beta})$ via projected supergradient ascent, exactly as described in section 3.1 on a generic active set $\mathcal{B}$. All the computations in the solver follow from keeping $\mathcal{B}=\emptyset$ in §3.1. We explicitly report them here for the reader's convenience.

Let us define the following shorthand for the primal coefficients:

$$\boldsymbol{f}_{\mathcal{M},k}(\boldsymbol{\alpha},\boldsymbol{\beta}_{\mathcal{M}}) = \left(\boldsymbol{\alpha}_k - W_{k+1}^T\boldsymbol{\alpha}_{k+1} - (\boldsymbol{\beta}_{k,0}+\boldsymbol{\beta}_{k,1}-W_{k+1}^T\boldsymbol{\beta}_{k,1})\right)$$

$$\boldsymbol{g}_{\mathcal{M},k}(\boldsymbol{\beta}_{\mathcal{M}}) = \boldsymbol{\beta}_{k,0}\odot\hat{\mathbf{u}}_k + \boldsymbol{\beta}_{k,1}\odot\hat{\mathbf{l}}_k.$$

The minimisation of the Lagrangian $\mathcal{L}_{\mathcal{M}}(\mathbf{x},\mathbf{z},\boldsymbol{\alpha},\boldsymbol{\beta})$ over the primals for $k\in[\![1,n-1]\!]$ is as follows:

$$\mathbf{x}_k^* = \mathbb{1}_{\boldsymbol{f}_{\mathcal{M},k}(\boldsymbol{\alpha},\boldsymbol{\beta}_{\mathcal{M}})\geq 0}\odot\hat{\mathbf{u}}_k + \mathbb{1}_{\boldsymbol{f}_{\mathcal{M},k}(\boldsymbol{\alpha},\boldsymbol{\beta}_{\mathcal{M}})<0}\odot\hat{\mathbf{l}}_k \qquad \mathbf{z}_k^* = \mathbb{1}_{\boldsymbol{g}_{\mathcal{M},k}(\boldsymbol{\beta}_{\mathcal{M}})\geq 0}\odot\mathbf{1} \tag{16}$$

For $k=0$, instead (assuming, as §3.1 that this can be done efficiently):

$$\mathbf{x}_0^* \in \operatorname*{argmin}_{\mathbf{x}_0} \ \boldsymbol{f}_{\mathcal{M},k}(\boldsymbol{\alpha},\boldsymbol{\beta}_{\mathcal{M}})^T\mathbf{x}_0 \qquad \text{s.t.} \quad \mathbf{x}_0\in\mathcal{C}. \tag{17}$$

The supergradient over the Big-M dual variables $\boldsymbol{\alpha},\boldsymbol{\beta}_{k,0},\boldsymbol{\beta}_{k,1}$ is computed exactly as in §3.1 and is again a subset of the supergradient of the full dual problem (6). We report it for completeness. For each $k\in[\![0,n-1]\!]$:

$$\nabla_{\boldsymbol{\alpha}_k}d(\boldsymbol{\alpha},\boldsymbol{\beta}) = W_k\mathbf{x}_{k-1}^* + \mathbf{b}_k - \mathbf{x}_k^*, \quad \nabla_{\boldsymbol{\beta}_{k,0}}d(\boldsymbol{\alpha},\boldsymbol{\beta}) = \mathbf{x}_k - \mathbf{z}_k\odot\hat{\mathbf{u}}_k,$$

$$\nabla_{\boldsymbol{\beta}_{k,1}}d(\boldsymbol{\alpha},\boldsymbol{\beta}) = \mathbf{x}_k - (W_k\mathbf{x}_{k-1}+\mathbf{b}_k) + (1-\mathbf{z}_k)\odot\hat{\mathbf{l}}_k. \tag{18}$$

## C DUAL DERIVATIONS

We now derive problem (6), the dual of the full relaxation by Anderson et al. (2020) described in equation (3). The Active Set (equation (8)) and Big-M duals (equation (15)) can be obtained by removing $\boldsymbol{\beta}_{k,I_k}\forall\ I_k\in\mathcal{E}_k\setminus\mathcal{B}_k$ and $\boldsymbol{\beta}_{k,I_k}\forall\ I_k\in\mathcal{E}_k$, respectively. We employ the following Lagrangian multipliers:

$$\mathbf{x}_k\geq\hat{\mathbf{x}}_k \Rightarrow \boldsymbol{\alpha}_k,$$

$$\mathbf{x}_k\leq\hat{\mathbf{u}}_k\odot\mathbf{z}_k \Rightarrow \boldsymbol{\beta}_{k,0},$$

$$\mathbf{x}_k\leq\hat{\mathbf{x}}_k-\hat{\mathbf{l}}_k\odot(1-\mathbf{z}_k) \Rightarrow \boldsymbol{\beta}_{k,1},$$

$$\mathbf{x}_k\leq\begin{pmatrix} (W_k\odot I_k)\mathbf{x}_{k-1}+\mathbf{z}_k\odot\mathbf{b}_k \\ -\left(W_k\odot I_k\odot\check{L}_{k-1}\right)\diamond(1-\mathbf{z}_k) \\ +\left(W_k\odot(1-I_k)\odot\check{U}_{k-1}\right)\diamond\mathbf{z}_k \end{pmatrix} \Rightarrow \boldsymbol{\beta}_{k,I_k},$$

and obtain, as a Lagrangian (using $\hat{\mathbf{x}}_k = W_k \mathbf{x}_{k-1} + \mathbf{b}_k$):

$$\mathcal{L}(\mathbf{x}, \mathbf{z}, \boldsymbol{\alpha}, \boldsymbol{\beta}) = \begin{bmatrix} \sum_{k=1}^{n-1} \boldsymbol{\alpha}_k^T (W_k \mathbf{x}_{k-1} + \mathbf{b}_k - \mathbf{x}_k) + \sum_{k=1}^{n-1} \boldsymbol{\beta}_{k,0}^T (\mathbf{x}_k - \mathbf{z}_k \odot \hat{\mathbf{u}}_k) \\ + \sum_{k=1}^{n-1} \sum_{I_k \in \mathcal{E}_k} \boldsymbol{\beta}_{k,I_k}^T \begin{pmatrix} (W_k \odot I_k \odot \check{L}_{k-1}) \diamond (1 - \mathbf{z}_k) - (W_k \odot I_k) \mathbf{x}_{k-1} \\ - \mathbf{b}_k \odot \mathbf{z}_k - (W_k \odot (1 - I_k) \odot \check{U}_{k-1}) \diamond \mathbf{z}_k + \mathbf{x}_k \end{pmatrix} \\ + \sum_{k=1}^{n-1} \boldsymbol{\beta}_{k,1}^T \left( \mathbf{x}_k - (W_k \mathbf{x}_{k-1} + \mathbf{b}_k) + (1 - \mathbf{z}_k) \odot \hat{\mathbf{l}}_k \right) + W_n \mathbf{x}_{n-1} + \mathbf{b}_n \end{bmatrix}$$

Let us use $\sum_{I_k}$ as shorthand for $\sum_{I_k \in \mathcal{E}_k \cup \{0,1\}}$. If we collect the terms with respect to the primal variables and employ dummy variables $\boldsymbol{\alpha}_0 = 0, \boldsymbol{\beta}_0 = 0, \boldsymbol{\alpha}_n = I, \boldsymbol{\beta}_n = 0$, we obtain:

$$\mathcal{L}(\mathbf{x}, \mathbf{z}, \boldsymbol{\alpha}, \boldsymbol{\beta}) = \begin{bmatrix} -\sum_{k=0}^{n-1} \begin{pmatrix} \boldsymbol{\alpha}_k - W_{k+1}^T \boldsymbol{\alpha}_{k+1} - \sum_{I_k} \boldsymbol{\beta}_{k,I_k} \\ + \sum_{I_{k+1}} (W_{k+1} \odot I_{k+1})^T \boldsymbol{\beta}_{k+1,I_{k+1}} \end{pmatrix}^T \mathbf{x}_k \\ -\sum_{k=1}^{n-1} \begin{pmatrix} \sum_{I_k \in \mathcal{E}_k} \boldsymbol{\beta}_{k,I_k} \odot \mathbf{b}_k + \boldsymbol{\beta}_{k,1} \odot \hat{\mathbf{l}}_k + \boldsymbol{\beta}_{k,0} \odot \hat{\mathbf{u}}_k \\ + \sum_{I_k \in \mathcal{E}_k} (W_k \odot I_k \odot \check{L}_{k-1}) \diamond \boldsymbol{\beta}_{k,I_k} \\ + \sum_{I_k \in \mathcal{E}_k} (W_k \odot (1 - I_k) \odot \check{U}_{k-1}) \diamond \boldsymbol{\beta}_{k,I_k} \end{pmatrix}^T \mathbf{z}_k \\ + \sum_{k=1}^{n-1} \mathbf{b}_k^T \boldsymbol{\alpha}_k + \sum_{k=1}^{n-1} \left( \sum_{I_k \in \mathcal{E}_k} (W_k \odot I_k \odot \check{L}_{k-1}) \square \boldsymbol{\beta}_{k,I_k} + \boldsymbol{\beta}_{k,1}^T (\hat{\mathbf{l}}_k - \mathbf{b}_k) \right) \end{bmatrix}$$

which corresponds to the form shown in problem (6).

# D  IMPLEMENTATION DETAILS FOR ACTIVE SET METHOD

From the high-level perspective, our Active Set solver proceeds by repeatedly solving modified instances of problem (6), where the exponential set $\mathcal{E}_k$ is replaced by a fixed (small) set of active variables $\mathcal{B}$. The full solver procedure is summarised in algorithm 2.

---

**Algorithm 2** Active Set solver

---

1:  **function** ACTIVESET_COMPUTE_BOUNDS($\{W_k, \mathbf{b}_k, \mathbf{l}_k, \mathbf{u}_k, \hat{\mathbf{l}}_k, \hat{\mathbf{u}}_k\}_{k=1..n}$)
2:      Initialise duals $\boldsymbol{\alpha}^0, \boldsymbol{\beta}_0^0, \boldsymbol{\beta}_1^0$ using Algorithm (1)
3:      Set $\boldsymbol{\beta}_{k,I_k} = 0, \forall\, I_k \in \mathcal{E}_k$
4:      $\mathcal{B} = \emptyset$
5:      **for** nb_additions **do**
6:          **for** $t \in [\![1, T-1]\!]$ **do**
7:              $\mathbf{x}^*, \mathbf{z}^* \in \operatorname{argmin}_{\mathbf{x},\mathbf{z}} \mathcal{L}_\mathcal{B}(\mathbf{x}, \mathbf{z}, \boldsymbol{\alpha}^t, \boldsymbol{\beta}_\mathcal{B}^t)$ using (9),(10)
8:              **if** $t \leq$ nb_vars_to_add **then**
9:                  For each layer $k$, add output of (4) called at $(\mathbf{x}^*, \mathbf{z}^*)$ to $\mathcal{B}_k$
10:             **end if**
11:             Compute supergradient using (11)
12:             $\boldsymbol{\alpha}^{t+1}, \boldsymbol{\beta}_\mathcal{B}^{t+1} \leftarrow$ Adam's update rule (Kingma & Ba, 2015)
13:             $\boldsymbol{\alpha}^{t+1}, \boldsymbol{\beta}_\mathcal{B}^{t+1} \leftarrow \max(\boldsymbol{\alpha}^{t+1}, 0), \max(\boldsymbol{\beta}_\mathcal{B}^{t+1}, 0)$      (dual projection)
14:         **end for**
15:     **end for**
16:     **return** $\min_{\mathbf{x},\mathbf{z}} \mathcal{L}_\mathcal{B}(\mathbf{x}, \mathbf{z}, \boldsymbol{\alpha}^T, \boldsymbol{\beta}_\mathcal{B}^T)$
17: **end function**

---

We conclude this section by proving the primal interpretation of the selection criterion for adding a new set of variables to $\mathcal{B}$.

## D.1  ACTIVE SET SELECTION CRITERION

We map a restricted set of dual variables $\boldsymbol{\beta}_\mathcal{B}$ to a set of dual variables $\boldsymbol{\beta}$ for the full dual (6) by setting variables not in the active set to 0: $\boldsymbol{\beta}_{\bar{\mathcal{B}}} = 0$, and $\boldsymbol{\beta} = \boldsymbol{\beta}_\mathcal{B} \cup \boldsymbol{\beta}_{\bar{\mathcal{B}}}$.

**Proposition.** *Let $\boldsymbol{\beta}_{k,I_k^*}$ be the set of dual variables maximising the corresponding entries of the supergradient of the full dual problem (6): $\boldsymbol{\beta}_{k,I_k^*} \in \operatorname{argmax}_{\boldsymbol{\beta}_{k,I_k}} \{\nabla_{\boldsymbol{\beta}_{k,I_k}} d(\boldsymbol{\alpha}, \boldsymbol{\beta})^T \mathbf{1}\}$. $\boldsymbol{\beta}_{k,I_k^*}$ represents the Lagrangian multipliers associated to the most violated constraints from $\mathcal{A}_k$ at $(\mathbf{x}^*, \mathbf{z}^*) \in \operatorname{argmin}_{\mathbf{x},\mathbf{z}} \mathcal{L}_\mathcal{B}(\mathbf{x}, \mathbf{z}, \boldsymbol{\alpha}, \boldsymbol{\beta}_\mathcal{B})$, the primal minimiser of the current restricted Lagrangian.*

*Proof.* In the following, $(\mathbf{x}^*, \mathbf{z}^*)$ denotes the points introduced in the statement. Recall the definition of $\nabla_{\boldsymbol{\beta}_{k,I_k}} d(\boldsymbol{\alpha}, \boldsymbol{\beta})$ in equation (9), which applies beyond the current active set:

$$\nabla_{\boldsymbol{\beta}_{k,I_k}} d(\boldsymbol{\alpha}, \boldsymbol{\beta}) = \left( \begin{array}{c} \mathbf{x}_k^* - (W_k \odot I_k)\mathbf{x}_{k-1}^* + \left(W_k \odot I_k \odot \check{L}_{k-1}\right) \diamond (1 - \mathbf{z}_k^*) \\ -\mathbf{z}_k^* \odot \mathbf{b}_k + \left(W_k \odot (1 - I_k) \odot \check{U}_{k-1}\right) \diamond \mathbf{z}_k^* \end{array} \right) \quad I_k \in \mathcal{E}_k.$$

We want to compute $I_k^* \in \operatorname{argmax}_{I_k} \{\nabla_{\boldsymbol{\beta}_{k,I_k}} d(\boldsymbol{\alpha}, \boldsymbol{\beta})^T \mathbf{1}\}$, that is:

$$I_k^* \in \operatorname*{argmax}_{I_k \in \mathcal{E}_k} \left( \begin{array}{c} \mathbf{x}_k^* - (W_k \odot I_k)\mathbf{x}_{k-1}^* + \left(W_k \odot I_k \odot \check{L}_{k-1}\right) \diamond (1 - \mathbf{z}_k^*) \\ -\mathbf{z}_k^* \odot \mathbf{b}_k + \left(W_k \odot (1 - I_k) \odot \check{U}_{k-1}\right) \diamond \mathbf{z}_k^* \end{array} \right)^T \mathbf{1}.$$

By removing the terms that do not depend on $I_k$, we obtain:

$$\max_{I_k \in \mathcal{E}_k} \left( \begin{array}{c} -(W_k \odot I_k)\mathbf{x}_{k-1}^* + \left(W_k \odot I_k \odot \check{L}_{k-1}\right) \diamond (1 - \mathbf{z}_k^*) \\ +\left(W_k \odot I_k \odot \check{U}_{k-1}\right) \diamond \mathbf{z}_k^* \end{array} \right)^T \mathbf{1}.$$

Let us denote the $i$-th row of $W_k$ and $I_k$ by $\boldsymbol{w}_{i,k}$ and $\boldsymbol{i}_{i,k}$, respectively, and define $\mathcal{E}_k[i] = 2^{\boldsymbol{w}_{i,k}} \setminus \{0, 1\}$. The optimisation decomposes over each such row: we thus focus on the optimisation problem for the supergradient's $i$-th entry. Collecting the mask, we get:

$$\max_{\boldsymbol{i}_{i,k} \in \mathcal{E}_k[i]} \sum_j \left( \left( (1 - \mathbf{z}_k^*[i]) \odot \check{L}_{k-1}[i,j] + \mathbf{z}_k^*[i] \odot \check{U}_{k-1}[i,j] - \mathbf{x}_{k-1}^*[i] \right) W_k[i,j] \right) I_k[i,j].$$

As the solution to the problem above is obtained by setting $I_k^*[i,j] = 1$ if its coefficient is positive and $I_k^*[i,j] = 0$ otherwise, we can see that the optimal $I_k$ corresponds to calling oracle (4) by Anderson et al. (2020) on $(\mathbf{x}^*, \mathbf{z}^*)$. Hence, in addition to being the mask associated to $\boldsymbol{\beta}_{k,I_k^*}$, the variable set maximising the supergradient, $I_k^*$ corresponds to the most violated constraint from $\mathcal{A}_k$ at $(\mathbf{x}^*, \mathbf{z}^*)$. $\qquad\square$

## E   INTERMEDIATE BOUNDS

A crucial quantity in both ReLU relaxations ($\mathcal{M}_k$ and $\mathcal{A}_k$) are intermediate pre-activation bounds $\hat{\mathbf{l}}_k, \hat{\mathbf{u}}_k$. In practice, they are computed by solving a relaxation $\mathcal{C}_k$ (which might be $\mathcal{M}_k$, $\mathcal{A}_k$, or something looser) of (1) over subsets of the network (Bunel et al., 2020a). For $\hat{\mathbf{l}}_i$, this means solving the following problem (separately, for each entry $\hat{\mathbf{l}}_i[j]$):

$$\begin{aligned} \min_{\mathbf{x}, \hat{\mathbf{x}}, \mathbf{z}} \quad & \hat{\mathbf{x}}_i[j] \\ \text{s.t.} \quad & \mathbf{x}_0 \in \mathcal{C} \\ & \hat{\mathbf{x}}_{k+1} = W_{k+1}\mathbf{x}_k + \mathbf{b}_{k+1}, \quad k \in [\![0, i-1]\!], \\ & (\mathbf{x}_k, \hat{\mathbf{x}}_k, \mathbf{z}_k) \in \mathcal{C}_k \qquad k \in [\![1, i-1]\!]. \end{aligned} \tag{19}$$

As (19) needs to be solved twice for each neuron (lower and upper bounds, changing the sign of the last layer's weights) rather than once as in (3), depending on the computational budget, $\mathcal{C}_k$ might be looser than the relaxation employed for the last layer bounds (in our case, $\mathcal{A}_k$). In all our experiments, we compute intermediate bounds as the tightest bounds between the method by Wong & Kolter (2018) and Interval Propagation (Gowal et al., 2018).

Once pre-activation bounds are available, post-activation bounds can be simply computed as $\mathbf{l}_k = \max(\hat{\mathbf{l}}_k, 0), \mathbf{u}_k = \max(\hat{\mathbf{u}}_k, 0)$.

## F   PRE-ACTIVATION BOUNDS IN $\mathcal{A}_k$

We now highlight the importance of an explicit treatment of pre-activation bounds in the context of the relaxation by Anderson et al. (2020). In §F.1 we will show through an example that, without a separate pre-activation bounds treatment, $\mathcal{A}_k$ could be looser than the less computationally expensive $\mathcal{M}_k$ relaxation. We then (§F.2) justify our specific pre-activation bounds treatment by extending the original proof by Anderson et al. (2020).

The original formulation by Anderson et al. (2020) is the following:

$$
\left.
\begin{aligned}
\mathbf{x}_k &\geq W_k \mathbf{x}_{k-1} + \mathbf{b}_k \\
\mathbf{x}_k &\leq \begin{pmatrix} (W_k \odot I_k)\,\mathbf{x}_{k-1} + \mathbf{z}_k \odot \mathbf{b}_k \\ -\left(W_k \odot I_k \odot \check{L}_{k-1}\right) \diamond (1 - \mathbf{z}_k) \\ +\left(W_k \odot (1 - I_k) \odot \check{U}_{k-1}\right) \diamond \mathbf{z}_k \end{pmatrix} \quad \forall I_k \in 2^{W_k} \\
(\mathbf{x}_k, \hat{\mathbf{x}}_k, \mathbf{z}_k) &\in [\mathbf{l}_k, \mathbf{u}_k] \times [\hat{\mathbf{l}}_k, \hat{\mathbf{u}}_k] \times [\mathbf{0},\ \mathbf{1}]
\end{aligned}
\right\} = \mathcal{A}'_k.
\tag{20}
$$

The difference with respect to $\mathcal{A}_k$ as defined in equation (3) exclusively lies in the treatment of pre-activation bounds. While $\mathcal{A}_k$ explicitly employs generic $\hat{\mathbf{l}}_k, \hat{\mathbf{u}}_k$ in the constraint set via $\mathcal{M}_k$, $\mathcal{A}'_k$ implicitly sets $\hat{\mathbf{l}}_k, \hat{\mathbf{u}}_k$ to the value dictated by interval propagation bounds (Gowal et al., 2018) via the constraints in $I_k = 0$ and $I_k = 1$ from the exponential family. In fact, setting $I_k = 0$ and $I_k = 1$, we obtain the following two constraints:

$$
\mathbf{x}_k \leq \hat{\mathbf{x}}_k - M_k^- \odot (1 - \mathbf{z}_k)
$$
$$
\mathbf{x}_k \leq M_k^+ \odot \mathbf{z}_k
$$
$$
\text{where:} \quad M_k^- := \min_{\mathbf{x}_{k-1} \in [\mathbf{l}_{k-1}, \mathbf{u}_{k-1}]} W_k^T \mathbf{x}_{k-1} + \mathbf{b}_k = W_k \odot \check{L}_{k-1} + \mathbf{b}_k
\tag{21}
$$
$$
M_k^+ := \max_{\mathbf{x}_{k-1} \in [\mathbf{l}_{k-1}, \mathbf{u}_{k-1}]} W_k^T \mathbf{x}_{k-1} + \mathbf{b}_k = W_k \odot \check{U}_{k-1} + \mathbf{b}_k
$$

which correspond to the upper bounding ReLU constraints in $\mathcal{M}_k$ if we set $\hat{\mathbf{l}}_k \rightarrow M_k^-$, $\hat{\mathbf{u}}_k \rightarrow M_k^+$. While $\hat{\mathbf{l}}_k, \hat{\mathbf{u}}_k$ are (potentially) computed solving an optimisation problem over the entire network (problem 19), the optimisation for $M_k^-, M_k^+$ involves only the layer before the current. Therefore, the constraints in (21) might be much looser than those in $\mathcal{M}_k$.

In practice, the effect of $\hat{\mathbf{l}}_k[i], \hat{\mathbf{u}}_k[i]$ on the resulting set is so significant that $\mathcal{M}_k$ might yield better bounds than $\mathcal{A}'_k$, even on very small networks. We now provide a simple example.

## F.1 MOTIVATING EXAMPLE

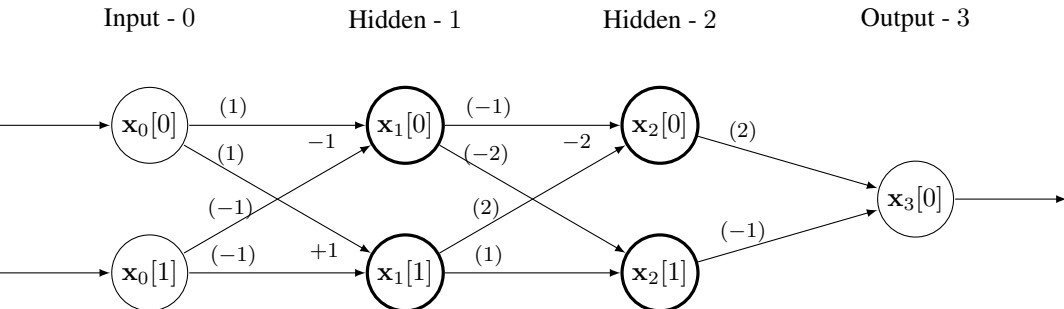

Figure 5: Example network architecture in which $\mathcal{M}_k \subset \mathcal{A}'_k$, with pre-activation bounds computed with $\mathcal{C}_k = \mathcal{M}_k$. For the bold nodes (the two hidden layers) a ReLU activation follows the linear function. The numbers between parentheses indicate multiplicative weights, the others additive biases (if any).

Figure 5 illustrates the network architecture. The size of the network is the minimal required to reproduce the phenomenon. $\mathcal{M}_k$ and $\mathcal{A}_k$ coincide for single-neuron layers (Anderson et al., 2020), and $\hat{\mathbf{l}}_k = M_k^-, \hat{\mathbf{u}}_k = M_k^+$ on the first hidden layer (hence, a second layer is needed).

Let us write the example network as a (not yet relaxed, as in problem (1)) optimization problem for the lower bound on the output node $\mathbf{x}_3$.

$$
\mathbf{l}_3 = \arg \min_{\mathbf{x}, \hat{\mathbf{x}}} \quad \begin{bmatrix} 2 & -1 \end{bmatrix} \mathbf{x}_2
\tag{22a}
$$

$$
\text{s.t.} \quad \mathbf{x}_0 \in [-1, 1]^2
\tag{22b}
$$

$$\hat{\mathbf{x}}_1 = \begin{bmatrix} 1 & -1 \\ 1 & -1 \end{bmatrix} \mathbf{x}_0 + \begin{bmatrix} -1 \\ 1 \end{bmatrix} \qquad \mathbf{x}_1 = \max(0, \hat{\mathbf{x}}_1) \tag{22c}$$

$$\hat{\mathbf{x}}_2 = \begin{bmatrix} -1 & 2 \\ -2 & 1 \end{bmatrix} \mathbf{x}_1 + \begin{bmatrix} -2 \\ 0 \end{bmatrix} \qquad \mathbf{x}_2 = \max(0, \hat{\mathbf{x}}_2) \tag{22d}$$

$$\mathbf{x}_3 = \begin{bmatrix} 2 & -1 \end{bmatrix} \mathbf{x}_2 \tag{22e}$$

Let us compute pre-activation bounds with $\mathcal{C}_k = \mathcal{M}_k$ (see problem (19)). For this network, the final output lower bound is tighter if the employed relaxation is $\mathcal{M}_k$ rather than $\mathcal{A}_k$ (hence, in this case, $\mathcal{M}_k \subset \mathcal{A}'_k$). Specifically: $\hat{\mathbf{l}}_{3,\mathcal{A}'_k} = -1.2857, \hat{\mathbf{l}}_{3,\mathcal{M}_k} = -1.2273$. In fact:

- In order to compute $\mathbf{l}_1$ and $\mathbf{u}_1$, the post-activation bounds of the first-layer, it suffices to solve a box-constrained linear program for $\hat{\mathbf{l}}_1$ and $\hat{\mathbf{u}}_1$, which at this layer coincide with interval propagation bounds, and to clip them to be non-negative. This yields $\mathbf{l}_1 = \begin{bmatrix} 0 & 0 \end{bmatrix}^T$, $\mathbf{u}_1 = \begin{bmatrix} 1 & 3 \end{bmatrix}^T$.

- Computing $M_2^+[1] = \max_{\mathbf{x}_1 \in [\mathbf{l}_1, \mathbf{u}_1]} \begin{bmatrix} -2 & 1 \end{bmatrix} \mathbf{x}_1 = 3$ we are assuming that $\mathbf{x}_1[0] = \mathbf{l}_1[0]$ and $\mathbf{x}_1[1] = \mathbf{u}_1[1]$. These two assignments are in practice conflicting, as they imply different values for $\mathbf{x}_0$. Specifically, $\mathbf{x}_1[1] = \mathbf{u}_1[1]$ requires $\mathbf{x}_0 = \begin{bmatrix} \mathbf{u}_0[0] & \mathbf{l}_0[1] \end{bmatrix} = \begin{bmatrix} 1 & -1 \end{bmatrix}$, but this would also imply $\mathbf{x}_1[0] = \mathbf{u}_1[0]$, yielding $\hat{\mathbf{x}}_2[1] = 1 \neq 3$.

  Therefore, explicitly solving a LP relaxation of the network for the value of $\hat{\mathbf{u}}_2[1]$ will tighten the bound. Using $\mathcal{M}_k$, the LP for this intermediate pre-activation bound is:

$$\hat{\mathbf{u}}_2[1] = \arg\min_{\mathbf{x},\hat{\mathbf{x}},\mathbf{z}} \begin{bmatrix} -2 & 1 \end{bmatrix} \mathbf{x}_1 \tag{23a}$$

$$\text{s.t.} \quad \mathbf{x}_0 \in [-1, 1]^2, \mathbf{z}_1 \in [0, 1]^2, \mathbf{x}_1 \in \mathbb{R}^2_{\geq 0} \tag{23b}$$

$$\hat{\mathbf{x}}_1 = \begin{bmatrix} 1 & -1 \\ 1 & -1 \end{bmatrix} \mathbf{x}_0 + \begin{bmatrix} -1 \\ 1 \end{bmatrix} \tag{23c}$$

$$\mathbf{x}_1 \geq \hat{\mathbf{x}}_1 \tag{23d}$$

$$\mathbf{x}_1 \leq \hat{\mathbf{u}}_1 \odot \mathbf{z}_1 = \begin{bmatrix} 1 \\ 3 \end{bmatrix} \odot \mathbf{z}_1 \tag{23e}$$

$$\mathbf{x}_1 \leq \hat{\mathbf{x}}_1 - \hat{\mathbf{l}}_1 \odot (1 - \mathbf{z}_1) = \hat{\mathbf{x}}_1 - \begin{bmatrix} -3 \\ -1 \end{bmatrix} \odot (1 - \mathbf{z}_1) \tag{23f}$$

  Yielding $\hat{\mathbf{u}}_2[1] = 2.25 < 3 = M_2^+[1]$. An analogous reasoning holds for $M_2^-[1]$ and $\hat{\mathbf{l}}_2[1]$.

- In $\mathcal{M}_k$, we therefore added the following two constraints:

$$\begin{aligned} \mathbf{x}_2[1] &\leq \hat{\mathbf{x}}_2[1] - \hat{\mathbf{l}}_2[1](1 - \mathbf{z}_2[1]) \\ \mathbf{x}_2[1] &\leq \hat{\mathbf{u}}_2[1]\mathbf{z}_2[1] \end{aligned} \tag{24}$$

  that in $\mathcal{A}'_k$ correspond to the weaker:

$$\begin{aligned} \mathbf{x}_2[1] &\leq \hat{\mathbf{x}}_2[1] - M_2^-[1](1 - \mathbf{z}_2[1]) \\ \mathbf{x}_2[1] &\leq M_2^+[1]\mathbf{z}_2[1] \end{aligned} \tag{25}$$

  As the last layer weight corresponding to $\mathbf{x}_2[1]$ is negative ($W_3[0, 1] = -1$), these constraints are going to influence the computation of $\hat{\mathbf{l}}_3$.

- In fact, the constraints in (24) are both active when optimizing for $\hat{\mathbf{l}}_{3,\mathcal{M}_k}$, whereas their counterparts for $\hat{\mathbf{l}}_{3,\mathcal{A}'_k}$ in (25) are not. The only active upper constraint at neuron $\mathbf{x}_2[1]$ for the Anderson relaxation is $\mathbf{x}_2[1] \leq \mathbf{x}_1[1]$, corresponding to the constraint from $\mathcal{A}'_2$ with $I_2[1, \cdot] = [0 \ 1]$. Evidently, its effect is not sufficient to counter-balance the effect of the tighter constraints (24) for $I_2[1, \cdot] = [1 \ 1]$ and $I_2[1, \cdot] = [0 \ 0]$, yielding a weaker lower bound for the network output.

F.2   DERIVATION OF $\mathcal{A}_k$

Having motivated an explicit pre-activation bounds treatment for the relaxation by Anderson et al. (2020), we now extend the original proof for $\mathcal{A}'_k$ (equation (20)) to obtain our formulation $\mathcal{A}_k$ (as defined in equation (3)). For simplicity, we will operate on a single neuron $\mathbf{x}_k[i]$.

A self-contained way to derive $\mathcal{A}'_k$ is by applying Fourier-Motzkin elimination on a standard MIP formulation referred to as the *multiple choice* formulation (Anderson et al., 2019), which is defined as follows:

$$\left.\begin{array}{l} (\mathbf{x}_{k-1}, \mathbf{x}_k[i]) = (\mathbf{x}^0_{k-1}, \mathbf{x}^0_k[i]) + (\mathbf{x}^1_{k-1}, \mathbf{x}^1_k[i]) \\ \mathbf{x}^0_k[i] = 0 \geq \boldsymbol{w}^T_{i,k} \mathbf{x}^0_{k-1} + \mathbf{b}_k[i](1 - \mathbf{z}_k[i]) \\ \mathbf{x}^1_k[i] = \boldsymbol{w}^T_{i,k} \mathbf{x}^1_{k-1} + \mathbf{b}_k[i]\mathbf{z}_k[i] \geq 0 \\ \mathbf{l}_{k-1}[i](1 - \mathbf{z}_k[i]) \leq \mathbf{x}^0_{k-1} \leq \mathbf{u}_{k-1}[i](1 - \mathbf{z}_k[i]) \\ \mathbf{l}_{k-1}[i]\mathbf{z}_k[i] \leq \mathbf{x}^1_{k-1} \leq \mathbf{u}_{k-1}[i]\mathbf{z}_k[i] \\ \mathbf{z}_k[i] \in [0, \ 1] \end{array}\right\} = \mathcal{S}'_{k,i} \qquad (26)$$

Where $\boldsymbol{w}_{i,k}$ denotes the $i$-th row of $W_k$, and $\mathbf{x}^1_{k-1}$ and $\mathbf{x}^0_{k-1}$ are copies of the previous layer variables. Applying (26) to the entire neural network results in a quadratic number of variables (relative to the number of neurons). The formulation can be obtained from well-known techniques from the MIP literature (Jeroslow, 1987) (it is the union of the two polyhedra for a passing and a blocking ReLU, operating in the space of $\mathbf{x}_{k-1}$). Anderson et al. (2019) show that $\mathcal{A}'_k = \text{Proj}_{\mathbf{x}_{k-1},\mathbf{x}_k,\mathbf{z}_k}(\mathcal{S}'_k)$.

If pre-activation bounds $\hat{\mathbf{l}}_k, \hat{\mathbf{u}}_k$ (computed as described in section E) are available, we can naturally add them to (26) as follows:

$$\left.\begin{array}{l} (\mathbf{x}_{k-1}, \mathbf{x}_k[i], \mathbf{z}_k[i]) \in \mathcal{S}'_{k,i} \\ \hat{\mathbf{l}}_k[i](1 - \mathbf{z}_k[i]) \leq \boldsymbol{w}^T_{i,k}\mathbf{x}^0_{k-1} + \mathbf{b}_k[i](1 - \mathbf{z}_k[i]) \leq \hat{\mathbf{u}}_k[i](1 - \mathbf{z}_k[i]) \\ \hat{\mathbf{l}}_k[i] \odot \mathbf{z}_k[i] \leq \boldsymbol{w}^T_{i,k}\mathbf{x}^1_{k-1} + \mathbf{b}_k[i]\mathbf{z}_k[i] \leq \hat{\mathbf{u}}_k[i]\mathbf{z}_k[i] \end{array}\right\} = \mathcal{S}_{k,i} \qquad (27)$$

We now prove that this formulation yields $\mathcal{A}_k$ when projecting out the copies of the activations.

**Proposition.** *Sets $\mathcal{S}_k$ from equation* (27) *and $\mathcal{A}_k$ from problem* (3) *are equivalent, in the sense that $\mathcal{A}_k = \text{Proj}_{\mathbf{x}_{k-1},\mathbf{x}_k,\mathbf{z}_k}(\mathcal{S}_k)$.*

*Proof.* In order to prove the equivalence, we will rely on Fourier-Motzkin elimination as in the original Anderson relaxation proof (Anderson et al., 2019). Going along the lines of the original proof, we start from (26) and eliminate $\mathbf{x}^1_{k-1}$, $\mathbf{x}^0_k[i]$ and $\mathbf{x}^1_k[i]$ exploiting the equalities. We then re-write all the inequalities as upper or lower bounds on $\mathbf{x}^0_{k-1}[0]$ in order to eliminate this variable. As Anderson et al. (2019), we assume $\boldsymbol{w}_{i,k}[0] > 0$. The proof generalizes by using $\check{L}$ and $\check{U}$ for $\boldsymbol{w}_{i,k}[0] < 0$, whereas if the coefficient is 0 the variable is easily eliminated. We get the following system:

$$\mathbf{x}^0_{k-1}[0] = \frac{1}{\boldsymbol{w}_{i,k}[0]}\left(\boldsymbol{w}^T_{i,k}\mathbf{x}_{k-1} - \sum_{j>1}\boldsymbol{w}_{i,k}[j]\mathbf{x}^0_{k-1}[j] + \mathbf{b}_k[i]\mathbf{z}_k[i] - \mathbf{x}_k[i]\right) \qquad (28a)$$

$$\mathbf{x}^0_{k-1}[0] \leq -\frac{1}{\boldsymbol{w}_{i,k}[0]}\left(\sum_{j>1}\boldsymbol{w}_{i,k}[j]\mathbf{x}^0_{k-1}[j] + \mathbf{b}_k[i](1 - \mathbf{z}_k[i])\right) \qquad (28b)$$

$$\mathbf{x}^0_{k-1}[0] \leq \frac{1}{\boldsymbol{w}_{i,k}[0]}\left(\boldsymbol{w}^T_{i,k}\mathbf{x}_{k-1} - \sum_{j>1}\boldsymbol{w}_{i,k}[j]\mathbf{x}^0_{k-1}[j] + \mathbf{b}_k[i]\mathbf{z}_k[i]\right) \qquad (28c)$$

$$\mathbf{l}_{k-1}[0](1 - \mathbf{z}_k[i]) \leq \mathbf{x}^0_{k-1}[0] \leq \mathbf{u}_{k-1}[0](1 - \mathbf{z}_k[i]) \qquad (28d)$$

$$\mathbf{x}^0_{k-1}[0] \leq \mathbf{x}_{k-1}[0] - \mathbf{l}_{k-1}[0]\mathbf{z}_k[i] \qquad (28e)$$

$$\mathbf{x}^0_{k-1}[0] \geq \mathbf{x}_{k-1}[0] - \mathbf{u}_{k-1}[0]\mathbf{z}_k[i] \qquad (28f)$$

$$\mathbf{x}_{k-1}^0[0] \leq \frac{1}{\boldsymbol{w}_{i,k}[0]} \left( \boldsymbol{w}_{i,k}^T \mathbf{x}_{k-1} - \sum_{j>1} \boldsymbol{w}_{i,k}[j]\mathbf{x}_{k-1}^0[j] + (\boldsymbol{b}_k[i] - \hat{\mathbf{l}}_k[i])\mathbf{z}_k[i] \right) \tag{28g}$$

$$\mathbf{x}_{k-1}^0[0] \geq \frac{1}{\boldsymbol{w}_{i,k}[0]} \left( \boldsymbol{w}_{i,k}^T \mathbf{x}_{k-1} - \sum_{j>1} \boldsymbol{w}_{i,k}[j]\mathbf{x}_{k-1}^0[j] + (\boldsymbol{b}_k[i] - \hat{\mathbf{u}}_k[i])\mathbf{z}_k[i] \right) \tag{28h}$$

$$\mathbf{x}_{k-1}^0[0] \geq \frac{1}{\boldsymbol{w}_{i,k}[0]} \left( (\hat{\mathbf{l}}_k[i] - \boldsymbol{b}_k[i])(1 - \mathbf{z}_k[i]) - \sum_{j>1} \boldsymbol{w}_{i,k}[j]\mathbf{x}_{k-1}^0[j] \right) \tag{28i}$$

$$\mathbf{x}_{k-1}^0[0] \leq \frac{1}{\boldsymbol{w}_{i,k}[0]} \left( (\hat{\mathbf{u}}_k[i] - \boldsymbol{b}_k[i])(1 - \mathbf{z}_k[i]) - \sum_{j>1} \boldsymbol{w}_{i,k}[j]\mathbf{x}_{k-1}^0[j] \right) \tag{28j}$$

where only inequalities (28g) to (28j) are not present in the original proof. We therefore focus on the part of the Fourier-Motzkin elimination that deals with them, and invite the reader to refer to Anderson et al. (2019) for the others. The combination of these new inequalities yields trivial constraints. For instance:

$$(28\text{i}) + (28\text{g}) \implies \hat{\mathbf{l}}_k[i] \leq \boldsymbol{w}_{i,k}^T \mathbf{x}_{k-1} + \boldsymbol{b}_k[i] = \hat{\mathbf{x}}_k[i] \tag{29}$$

which holds by the definition of pre-activation bounds.

Let us recall that $\mathbf{x}_k[i] \geq 0$ and $\mathbf{x}_k[i] \geq \hat{\mathbf{x}}_k[i]$, the latter constraint resulting from (28a) + (28b). Then, it can be easily verified that the only combinations of interest (i.e., those that do not result in constraints that are obvious by definition or are implied by other constraints) are those containing the equality (28a). In particular, combining inequalities (28g) to (28j) with inequalities (28d) to (28f) generates constraints that are (after algebraic manipulations) superfluous with respect to those in (30). We are now ready to show the system resulting from the elimination:

$$\mathbf{x}_k[i] \geq 0 \tag{30a}$$

$$\mathbf{x}_k[i] \geq \hat{\mathbf{x}}_k[i] \tag{30b}$$

$$\mathbf{x}_k[i] \leq \boldsymbol{w}_{i,k}[0]\mathbf{x}_{k-1}[0] - \boldsymbol{w}_{i,k}[0]\mathbf{l}_{k-1}[0](1 - \mathbf{z}_k[i]) + \sum_{j>1} \boldsymbol{w}_{i,k}[j]\mathbf{x}_{k-1}^0[j] + \boldsymbol{b}_k[i]\mathbf{z}_k[i] \tag{30c}$$

$$\mathbf{x}_k[i] \leq \boldsymbol{w}_{i,k}[0]\mathbf{u}_{k-1}[0]\mathbf{z}_k[i] + \sum_{j>1} \boldsymbol{w}_{i,k}[j]\mathbf{x}_{k-1}^0[j] + \boldsymbol{b}_k[i]\mathbf{z}_k[i] \tag{30d}$$

$$\mathbf{x}_k[i] \geq \boldsymbol{w}_{i,k}[0]\mathbf{x}_{k-1}[0] - \boldsymbol{w}_{i,k}[0]\mathbf{u}_{k-1}[0](1 - \mathbf{z}_k[i]) + \sum_{j>1} \boldsymbol{w}_{i,k}[j]\mathbf{x}_{k-1}^0[j] + \boldsymbol{b}_k[i]\mathbf{z}_k[i] \tag{30e}$$

$$\mathbf{x}_k[i] \geq \boldsymbol{w}_{i,k}[0]\mathbf{l}_{k-1}[0]\mathbf{z}_k[i] + \sum_{j>1} \boldsymbol{w}_{i,k}[j]\mathbf{x}_{k-1}^0[j] + \boldsymbol{b}_k[i]\mathbf{z}_k[i] \tag{30f}$$

$$\mathbf{l}_{k-1}[0] \leq \mathbf{x}_k[i] \leq \mathbf{u}_{k-1}[0] \tag{30g}$$

$$\mathbf{x}_k[i] \geq \hat{\mathbf{l}}_k[i]\mathbf{z}_k[i] \tag{30h}$$

$$\mathbf{x}_k[i] \leq \hat{\mathbf{u}}_k[i]\mathbf{z}_k[i] \tag{30i}$$

$$\mathbf{x}_k[i] \leq \hat{\mathbf{x}}_k[i] - \hat{\mathbf{l}}_k[i](1 - \mathbf{z}_k[i]) \tag{30j}$$

$$\mathbf{x}_k[i] \geq \hat{\mathbf{x}}_k[i] - \hat{\mathbf{u}}_k[i](1 - \mathbf{z}_k[i]) \tag{30k}$$

Constraints from (30a) to (30g) are those resulting from the original derivation of $\mathcal{A}'_k$ (see (Anderson et al., 2019)). The others result from the inclusion of pre-activation bounds in (27). Of these, (30h) is implied by (30a) if $\hat{\mathbf{l}}_k[i] \leq 0$ and by the definition of pre-activation bounds (together with (30b)) if $\hat{\mathbf{l}}_k[i] > 0$. Analogously, (30k) is implied by (30b) if $\hat{\mathbf{u}}_k[i] \geq 0$ and by (30a) otherwise.

By noting that no auxiliary variable is left in (30i) and in (30j), we can conclude that these will not be affected by the remaining part of the elimination procedure. Therefore, the rest of the proof

(the elimination of $\mathbf{x}_{k-1}^0[1]$, $\mathbf{x}_{k-1}^0[2]$, ... ) proceeds as in (Anderson et al., 2019), leading to $\mathcal{A}_{k,i}$. Repeating the proof for each neuron $i$ at layer $k$, we get $\mathcal{A}_k = \mathrm{Proj}_{\mathbf{x}_{k-1}, \mathbf{x}_k, \mathbf{z}_k}(\mathcal{S}_k)$.

$\square$

## G    MASKED FORWARD AND BACKWARD PASSES

Crucial to the practical efficiency of our solvers is to represent the various operations as standard forward/backward passes over a neural network. This way, we can leverage the engineering efforts behind popular deep learning frameworks such as PyTorch (Paszke et al., 2017). While this can be trivially done for the Big-M solver (appendix B), the Active Set method (§3.1) requires a specialised operator that we call "masked" forward/backward pass. Here, we provide the details to our implementation.

The masked forward and backward passes respectively take the following forms: $(W_k \odot I_k)\, \mathbf{x}_k$, $(W_k \odot I_k)^T \mathbf{x}_{k+1}$ and they are needed when dealing with the exponential family of constraints from the relaxation by Anderson et al. (2020). A crucial property of the operator is that the $I_k$ mask may take on a different value for each input/output combination. While this is straightforward to implement for fully connected layers, we need to be more careful when handling convolutional operators, which rely on re-applying the same weights (kernel) to many different parts of the image. A naive solution is to convert convolutions into equivalent linear operators, but this has a high cost in terms of performance, as it involves much redundancy.

A convolutional operator can be represented via a matrix-matrix multiplication if the input is *unfolded* and the filter is appropriately reshaped. The multiplication output can then be reshaped to the correct convolutional output shape. Given a filter $w \in \mathbb{R}^{c \times k_1 \times k_2}$, an input $\mathbf{x} \in \mathbb{R}^{i_1 \times i_2 \times i_3}$ and the convolutional output $\mathrm{conv}_w(\mathbf{x}) = \mathbf{y} \in \mathbb{R}^{c \times o_2 \times o_3}$, we need the following definitions:

$$\begin{aligned}
[\cdot]_{\mathcal{I},\mathcal{O}} &: \mathcal{I} \to \mathcal{O} \\
\{\cdot\}_j &: \mathbb{R}^{d_1 \times \cdots \times d_n} \to \mathbb{R}^{d_1 \times \cdots \times d_{j-1} \times d_{j+1} \times \cdots \times d_n} \\
\mathtt{unfold}_w(\cdot) &: \mathbb{R}^{i_1 \times i_2 \times i_3} \to \mathbb{R}^{k_1 k_2 \times o_2 o_3} \\
\mathtt{fold}_w(\cdot) &: \mathbb{R}^{k_1 k_2 \times o_2 o_3} \to \mathbb{R}^{i_1 \times i_2 \times i_3}
\end{aligned} \tag{31}$$

where the brackets simply reshape the vector from shape $\mathcal{I}$ to $\mathcal{O}$, while the braces sum over the $j$-th dimension. $\mathtt{unfold}$ decomposes the input image into the (possibly overlapping) $o_2 o_3$ blocks the sliding kernel operates on, taking padding and striding into account. $\mathtt{fold}$ brings the output of $\mathtt{unfold}$ to the original input space.

Let us define the following reshaped versions of the filter and the convolutional output:

$$\begin{aligned}
W_R &= [w]_{\mathbb{R}^{c \times k_1 \times k_2}, \mathbb{R}^{c \times k_1 k_2}} \\
\mathbf{y}_R &= [\mathbf{y}]_{\mathbb{R}^{c \times o_2 \times o_3}, \mathbb{R}^{c \times o_2 o_3}}
\end{aligned}$$

The standard forward/backward convolution (neglecting the convolutional bias, which can be added at the end of the forward pass) can then be written as:

$$\begin{aligned}
\mathrm{conv}_w(\mathbf{x}) &= [W_R\, \mathtt{unfold}_w(\mathbf{x})]_{\mathbb{R}^{c \times o_2 o_3}, \mathbb{R}^{c \times o_2 \times o_3}} \\
\mathtt{back\_conv}_w(\mathbf{y}) &= \mathtt{fold}_w(W_R^T\, \mathbf{y}_R).
\end{aligned} \tag{32}$$

We need to mask the convolution with a different mask for each input-output pair. This means employing a mask $I \in \mathbb{R}^{c \times k_1 k_2 \times o_2 o_3}$. Therefore, assuming vectors are broadcast to the correct output shape[4], we can write the masked forward and backward passes as:

$$\begin{aligned}
\mathrm{conv}_{w,I}(\mathbf{x}) &= [\{(W_R \odot I \odot \mathtt{unfold}_w(\mathbf{x})\}_2]_{\mathbb{R}^{c \times o_2 o_3}, \mathbb{R}^{c \times o_2 \times o_3}} \\
\mathtt{back\_conv}_{w,I}(\mathbf{y}) &= \mathtt{fold}_w(\{W_R \odot I \odot \mathbf{y}_R\}_1).
\end{aligned} \tag{33}$$

Owing to the avoided redundancy with respect to the equivalent linear operation (e.g., copying of the kernel matrix, zero-padding in the linear weight matrix), this implementation of the masked for-

---

[4]if we want to perform an element-wise product $\mathbf{a} \odot \mathbf{b}$ between $\mathbf{a} \in \mathbb{R}^{d_1 \times d_2 \times d_3}$ and $\mathbf{b} \in \mathbb{R}^{d_1 \times d_3}$, the operation is implicitly performed as $\mathbf{a} \odot \mathbf{b}'$, where $\mathbf{b}' \in \mathbb{R}^{d_1 \times d_2 \times d_3}$ is an extended version of $\mathbf{b}$ obtained by copying along the missing dimension.

ward/backward pass reduces both the memory footprint and the number of floating point operations (FLOPs) associated to the passes computations by a factor $(i_1 i_2 i_3)/(k_1 k_2)$. In practice, this ratio might be significant: on the incomplete verification networks §5.1) it ranges from 16 to 64.

## H   STRATIFIED BOUNDING FOR BRANCH AND BOUND

As seen in the complete verification results in Figure 3 (§ 5.2), the use of a tighter bounding algorithm results in the verification of a larger number of properties. In general, tighter bounds come at a larger computational cost, which might negatively affect performance on "easy" verification properties, where a small number of domain splits with loose bounds suffices to verify the property (hence, the tightness is not needed). As a general complete verification system needs to be efficient a priori, on both easy and hard properties, we employ a *stratified* bounding system for use within Branch and Bound-like complete verification algorithms.

We design a set of heuristics to determine whether the children (i.e., the subproblems arising after splitting on it) of a given subproblem will require tight bounds or not. The graph describing parent-child relations between sub-problems is referred to as the BaB search tree. Given a problem $p$, we individually mark it as difficult (with all its future offspring) if it meets *all* the conditions in the following set:

- The lower bound on the minimum over the subproblem, $p_{LB}$, should be relatively "close" to the decision threshold (0, in the standard form by Bunel et al. (2018)). That is, $p_{LB} \leq c_{LB}$. We argue that tighter bounds are worth computing only if they are likely to result in a crossing of the decision threshold, thus cutting away a part of the BaB search tree.

- The depth $p_{depth}$ in the BaB search tree should not be below a certain threshold $c_{depth}$. This is a way to ensure a certain number of splits is performed before adopting tighter bounds.

- The running average of the lower bound improvement from parent to child $p_{impr}$ should fall below a threshold $c_{impr}$. This reflects the idea that if splitting seems to be effective on a given section of the BaB tree, it is perhaps wiser to invest computational budget in more splits (with cheaper bounding computations, more splits can be performed in a given time) than tighter bounds. Empirically, this is the criterion with the largest effect on performance.

Additionally, we do not employ tighter bounds unless the verification problem itself (in addition to the individual subproblems) is marked as "hard". We do so when the size of the un-pruned domains reaches a certain threshold $c_{hard}$.

The set of criteria above needs to address the following decision trade-off: one should postpone the use of tighter bounds until sure the difficulty of the verification task requires it. At the same time, if the switch to tighter bounds is excessively delayed, one might end up computing tight bounds on a large number of subproblems (as sections of the BaB tree were not pruned in time), hurting performance for harder tasks. In practice, we set the criteria as follows for all our experiments: $c_{depth} = 0$, $c_{LB} = 0.5$, $c_{impr} = 10^{-1}$, $c_{hard} = 200$. As seen in Figure 3, this resulted in a reasonable trade-off between losing efficiency on the harder properties (wide model) and gaining it on the easier ones (base and deep models).

# I  EXPERIMENTAL APPENDIX

We conclude the appendix by presenting supplementary experiments and additional details with respect to the presentation in the main paper.

## I.1  EXPERIMENTAL SETTING, HYPER-PARAMETERS

All the experiments and bounding computations (including intermediate bounds) were run on a single Nvidia Titan Xp GPU, except Gurobi-based methods and "Active Set CPU". These were run on i7-6850K CPUs, utilising 4 cores for the incomplete verification experiments, and 6 cores for the more demanding complete verification experiments. The experiments were run under Ubuntu 16.04.2 LTS. Complete verification results for ERAN, the method by Singh et al. (2020), are taken from the recent VNN-COMP competition (VNN-COMP, 2020). These were executed by Singh et al. (2020) on a 2.6 GHz Intel Xeon CPU E5-2690 with 512 GB of main memory, utilising 14 cores.

Gurobi-based methods make use of LP incrementalism (warm-starting) when possible. In the experiments of §5.1, where each image involves the computation of 9 different output upper bounds, we warm-start each LP from the LP of the previous neuron. For "Gurobi 1 cut", which involves two LPs per neuron, we first solve all Big-M LPs, then proceed with the LPs containing a single cut.

Hyper-parameter tuning for incomplete verification was done on a small subset of the CIFAR-10 test set, on the SGD-trained network from Figures 1, 2. BDD+ is run with the hyper-parameters found by Bunel et al. (2020a) on the same datasets, for both incomplete and complete verification. For all supergradient-based methods (Big-M, Active Set), we employed the Adam update rule (Kingma & Ba, 2015), which showed stronger empirical convergence. For Big-M, replicating the findings by Bunel et al. (2020a) on their supergradient method, we linearly decrease the step size from $10^{-2}$ to $10^{-4}$. Active Set is initialized with 500 Big-M iterations, after which the step size is reset and linearly scaled from $10^{-3}$ to $10^{-6}$. We found the addition of variables to the active set to be effective before convergence: we add variables every 450 iterations, without re-scaling the step size again. Every addition consists of 2 new variables (see pseudo-code in appendix D), which was found to be a good compromise between fast bound improvement and computational cost. On complete verification, we re-employed the same hyper-parameters for both Big-M and Active Set, except the number of iterations. For Big-M, this was tuned to employ roughly the same time per bounding computation as BDD+.

As the complete verification problem is formulated as a minimisation (as in problem (1)), in Branch and Bound we need a lower and an upper bound to the minimum of each sub-problem. The lower bound is computed by running the bounding algorithm, while the upper bound on the minimum is obtained by evaluating the network at the last primal output by the bounding algorithm in the lower bound computation (running the bounding algorithm to get an upper bound would result in a much looser bound, as it would imply having an upper bound on a version of problem (1) with maximisation instead of minimisation).

## I.2  DATASET DETAILS

We now provide further details on the employed datasets. For incomplete verification, the architecture was introduced by Wong & Kolter (2018) and re-employed by Bunel et al. (2020a). It corresponds to the "Wide" complete verification architecture, found in Table 2 . Due to the additional computational cost of bounds obtained via the tighter relaxation (3), we restricted the experiments to the first 3450 CIFAR-10 test set images for the experiments on the SGD-trained network (Figures 1, 2), and to the first 4513 images for the network trained via the method by Madry et al. (2018) (Figures 6, 7).

For complete verification, we employed a subset of the adversarial robustness dataset presented by Lu & Kumar (2020) and used by Bunel et al. (2020a), where the set of properties per network has been restricted to 100. The dataset provides, for a subset of the CIFAR-10 test set, a verification radius $\epsilon_{ver}$ defining the small region over which to look for adversarial examples (input points for which the output of the network is not the correct class) and a (randomly sampled) non-correct class to verify against. The verification problem is formulated as the search for an adversarial example, carried out by minimizing the difference between the ground truth logit and the target logit. If the minimum is positive, we have not succeeded in finding a counter-example, and the network is robust. The $\epsilon_{ver}$ radius was tuned to meet a certain "problem difficulty" via binary search, employing

| Network Name | No. of Properties | Network Architecture |
|---|---|---|
| BASE Model | 100 | Conv2d(3,8,4, stride=2, padding=1)
Conv2d(8,16,4, stride=2, padding=1)
linear layer of 100 hidden units
linear layer of 10 hidden units
(Total ReLU activation units: 3172) |
| WIDE | 100 | Conv2d(3,16,4, stride=2, padding=1)
Conv2d(16,32,4, stride=2, padding=1)
linear layer of 100 hidden units
linear layer of 10 hidden units
(Total ReLU activation units: 6244) |
| DEEP | 100 | Conv2d(3,8,4, stride=2, padding=1)
Conv2d(8,8,3, stride=1, padding=1)
Conv2d(8,8,3, stride=1, padding=1)
Conv2d(8,8,4, stride=2, padding=1)
linear layer of 100 hidden units
linear layer of 10 hidden units
(Total ReLU activation units: 6756) |

Table 2: For each complete verification experiment, the network architecture used and the number of verification properties tested, a subset of the dataset by Lu & Kumar (2020). Each layer but the last is followed by a ReLU activation function.

a Gurobi-based bounding algorithm (Lu & Kumar, 2020). The networks are robust on all the properties we employed. Three different network architectures of different sizes are used. A "Base" network with 3172 ReLU activations, and two networks with roughly twice as many activations: a "Deep" network, and a "Wide" network. Details can be found in Table 2.

### I.3    Adversarially-Trained Incomplete Verification

Figure 6: Upper plot: distribution of runtime in seconds. Lower plot: difference with the bounds obtained by Gurobi with a cut from $\mathcal{A}_k$ per neuron; higher is better. Results for the network adversarially trained with the method by Madry et al. (2018), from Bunel et al. (2020a). The width at a given value represents the proportion of problems for which this is the result.

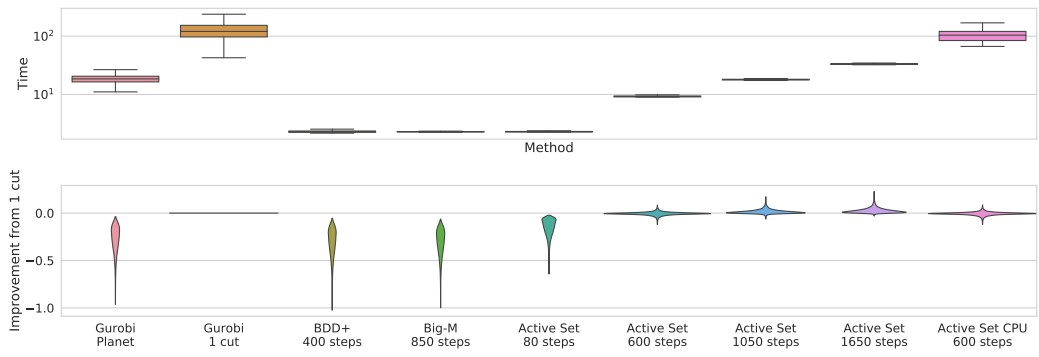

In addition to the SGD-trained network in §5.1, we now present results relative to the same architecture, trained with the adversarial training method by Madry et al. (2018) for robustness to perturbations of $\epsilon_{train} = 8/255$. Each adversarial sample for the training was obtained using 50 steps of projected gradient descent. For this network, we measure the robustness margin to perturbations with $\epsilon_{ver} = 12/255$.

Figures 6, 7 confirm most of the observations carried out for the SGD-trained network in §5.1, with fewer variability around the bounds returned by Gurobi cut. Big-M is competitive with BDD+, and switching to Active Set after $500$ iterations results in much better bounds in the same time. Increasing the computational budget for Active Set still results in better bounds than Gurobi cut in

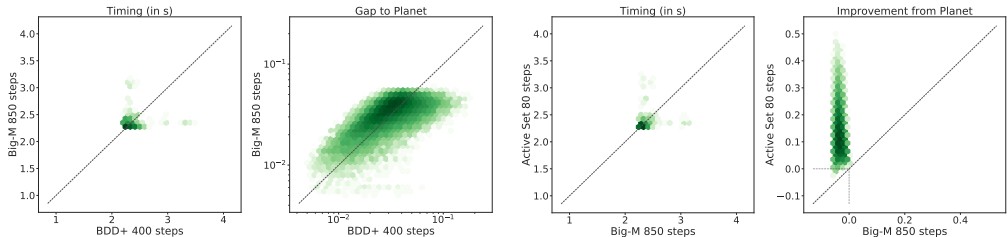

(a) Comparison of runtime (left) and gap to Gurobi Planet bounds. For the latter, lower is better.

(b) Comparison of runtime (left) and improvement from the Gurobi Planet bounds. For the latter, higher is better.

Figure 7: Pointwise comparison for a subset of the methods on the data presented in Figure 6. Darker colour shades mean higher point density (on a logarithmic scale). The oblique dotted line corresponds to the equality.

a fraction of its running time, even though the performance gap is on average smaller than on the SGD-trained network.

### I.4 SENSITIVITY TO SELECTION CRITERION AND FREQUENCY

In section 3.2, we describe how to iteratively modify $\mathcal{B}$, the active set of dual variables on which our Active Set solver operates. In short, Active Set adds the variables corresponding to the output

Figure 8: Upper plot: distribution of runtime in seconds. Lower plot: difference with the bounds obtained by Gurobi with a cut from $\mathcal{A}_k$ per neuron; higher is better. Results for the SGD-trained network from Bunel et al. (2020a). The width at a given value represents the proportion of problems for which this is the result. Sensitivity of Active Set to selection criterion (see §3.2).

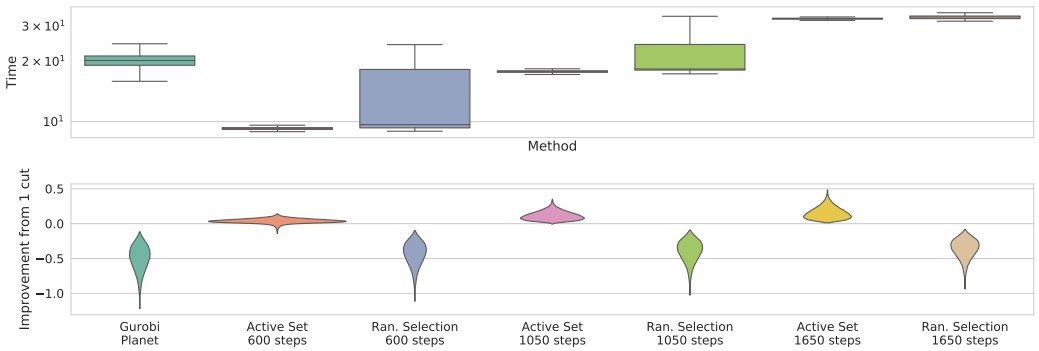

Figure 9: Upper plot: distribution of runtime in seconds. Lower plot: difference with the bounds obtained by Gurobi with a cut from $\mathcal{A}_k$ per neuron; higher is better. Results for the SGD-trained network from Bunel et al. (2020a). The width at a given value represents the proportion of problems for which this is the result. Sensitivity of Active Set to variable addition frequency $\omega$, with the selection criterion presented in §3.2.

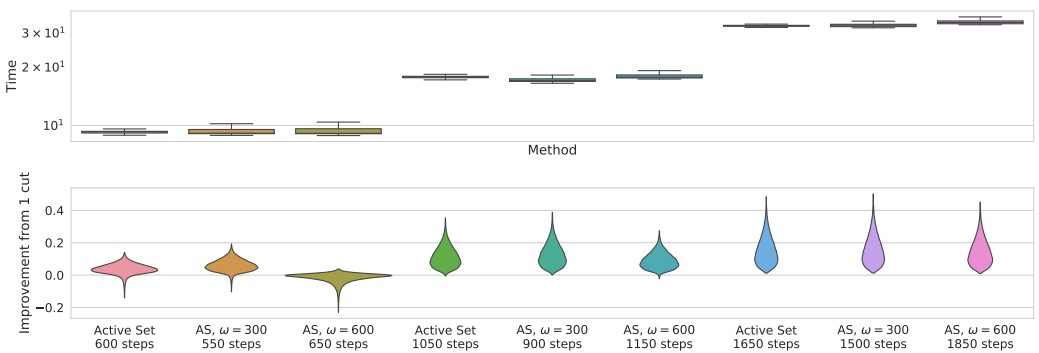

of oracle (4) invoked at the primal minimiser of $\mathcal{L}_{\mathcal{B}}(\mathbf{x}, \mathbf{z}, \boldsymbol{\alpha}, \boldsymbol{\beta}_{\mathcal{B}})$, at a fixed frequency $\omega$. We now investigate the empirical sensitivity of Active Set to both the selection criterion and the frequency of addition.

We test against **Ran. Selection**, a version of Active Set for which the variables to add are selected at random by uniformly sampling from the binary $I_k$ masks. As expected, Figure 8 shows that a good selection criterion is key to the efficiency of Active Set. In fact, random variable selection only marginally improves upon the Planet relaxation bounds, whereas the improvement becomes significant with our selection criterion from §3.2.

In addition, we investigate the sensitivity of Active Set (AS) to variable addition frequency $\omega$. In order to do so, we cap the maximum number of cuts at 7 for all runs, and vary $\omega$ while keeping the time budget fixed (we test on three different time budgets). Figure 9 compares the results for $\omega = 450$ (Active Set), which were presented in §5.1, with the bounds obtained by setting $\omega = 300$ and $\omega = 600$ (respectively **AS $\omega = 300$** and **AS $\omega = 600$**). Our solver proves to be relatively robust to $\omega$ across all the three budgets, with the difference in obtained bounds decreasing with the number of iterations. Moreover, early cut addition tends to yield better bounds in the same time, suggesting that our selection criterion is effective before subproblem convergence.

### I.5 MNIST INCOMPLETE VERIFICATION

We conclude the experimental appendix by presenting incomplete verification results (the experimental setup mirrors the one employed in section 5.1 and appendix I.3) on the MNIST dataset (Le-Cun et al., 1998).

We report results on the "wide" MNIST network from Lu & Kumar (2020), whose architecture is identical to the "wide" network in Table 2 except for the first layer, which has only one input channel to reflect the MNIST specification (the total number of ReLU activation units is 4804). As those employed for the complete verification experiments (§5.2), and differently from the incomplete

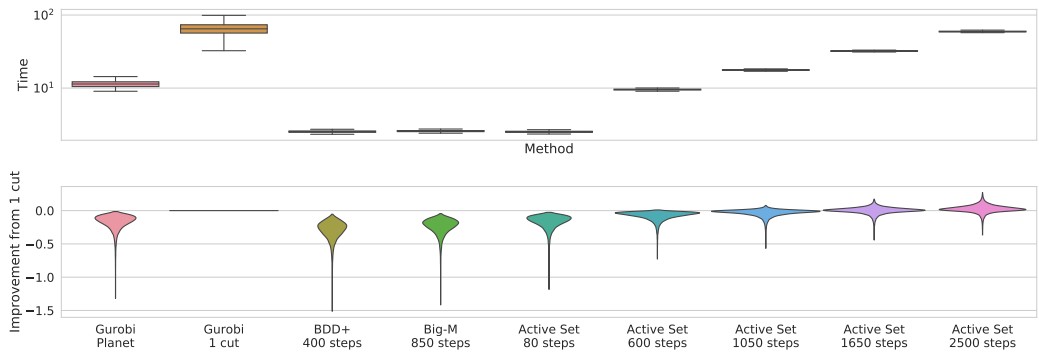

Figure 10: Upper plot: distribution of runtime in seconds. Lower plot: difference with the bounds obtained by Gurobi with a cut from $\mathcal{A}_k$ per neuron; higher is better. MNIST results for a network adversarially trained with the method by Wong & Kolter (2018), from Lu & Kumar (2020). The width at a given value represents the proportion of problems for which this is the result.

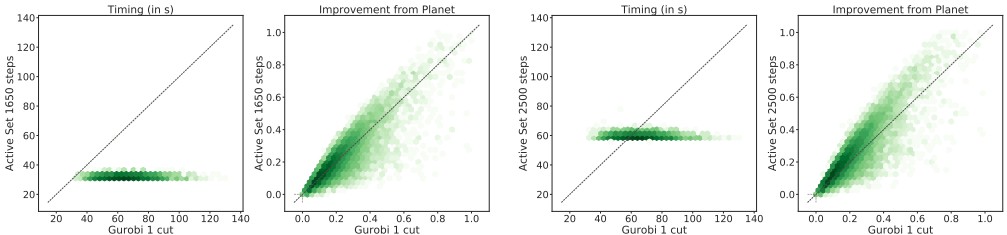

Figure 11: Pointwise comparison for a subset of the methods on the data presented in Figure 10. Comparison of runtime (left) and improvement from the Gurobi Planet bounds. For the latter, higher is better. Darker colour shades mean higher point density (on a logarithmic scale). The oblique dotted line corresponds to the equality.

verification experiments in section 5.1 and appendix I.3, the network was adversarially trained with the method by Wong & Kolter (2018). We compute the robustness margin to $\epsilon_{ver} = 0.15$ on the first 2960 images of the MNIST test set. All hyper-parameters are kept to the values employed for the CIFAR-10 networks, except the Big-M step size, which was linearly decreased from $10^{-1}$ to $10^{-3}$, and the weight of the proximal terms for BDD+, which was linearly increased from 1 to 50.

As seen on the CIFAR-10 networks, Figures 10, 11 show that Active Set yields comparable or better bounds than Gurobi 1 cut in less average runtime. However, more iterations are required to reach the same relative bound improvement over Gurobi 1 cut (2500 as opposed to 600 in Figures 1, 6). Furthermore, the smaller average gap between the bounds of Gurobi Planet and Gurobi 1 cut (especially with respect to Figure 1) suggests that the relaxation by Anderson et al. (2020) is less effective on this MNIST benchmark.

