# OpenReview forum: "Scaling the Convex Barrier with Active Sets"
_ICLR.cc/2021/Conference — ICLR 2021 Poster_

### Official Review · AnonReviewer4 · 2020-10-26
**A new solver employing a recently introduced formulation**

**Rating:** 6
**Confidence:** 1

**Review:**

The paper focuses on the notion of dual solvers, which are solvers that have the ability to prove as well as disprove that a specified property of neural networks holds over all inputs in a specified domain. Unsurprisingly, the bottle neck for this approach is the computation of lower bounds, which can be translated into a non-linear (and NP-hard) optimization problem. Dual solvers typically instead attempt to solve a linear formulation of the problem, and recently Anderson et al. introduced a new MIP formulation for the problem which is more accurate than previously used formulations at the cost of potentially requiring a significantly larger number of constraints.

In this work, the authors introduce a dual solver called "Active Set" that is designed to solve Anderson's tighter formulation, with the aim of achieving increased accuracy while keeping the computational costs low. The solver seems correct, even though I couldn't verify the details; generally speaking, it operates by starting with the easier but less accurate formulation that is classically used, and making it more precise by the gradual introduction of (an "active set" of) variables from a modified version of Anderson et al's more accurate formulation. Experimental evaluations show that the new solver can be used both for incomplete verification and complete verification; in both cases, it is possible to use Active Set to obtain an implementation that can either achieve higher accuracy (for incomplete verification) or avoid a timeout for a greater percentage of instances (for complete verification).

The paper is well written. The preliminaries and writeup in general are clearly aimed at experts with some background knowledge in the given area (which prevented me from verifying the technical details), but I believe that it fits well within the scope of ICLR. The main contribution is the design of a new solver and its experimental evaluation; the theoretical contribution is negligible. It is also easy to imagine that there could be many alternate ways one could use Anderson et al.'s recent formulation to improve the state of the art - in that sense, the results achieved by the authors are not surprising. But that does not mean that there were no challenges left to overcome, and the experimental evaluations seem to be reasonable. So I think that the paper's contribution is sufficient to warrant a presentation at ICLR.

Question:
-Are there other "standard" datasets that could be considered instead of CIFAR-10, and would it be difficult to also use these for experimental evaluations?

Minor remarks:
-page 2: "if more compute budget is..."  should be "if a larger computational budget is..."

Post-rebuttal comment:
I acknowledge having read the authors' response and I have also glanced over the updated version of the paper.

---

> ### Author Response · Authors · 2020-11-17
> **We thank the reviewer for their feedback**
>
> We would like to thank the reviewer for their positive comments and feedback.
>
> **Potential of Anderson et al.**
> While we agree that the work by Anderson et al. has great potential for piecewise-linear neural network verification, we would like to stress that the formulation per se (in its primal form) does not improve upon the state-of-the-art for complete verification. This is visible in Figure 3 by comparing the performance of “G. Planet + G. 1 cut BaBSR” and the MIP implementation with BDD+.

---

> > ### Author Response · Authors · 2020-11-23
> > **We additionally provide MNIST experiments**
> >
> > **Additional experiments**
> > We thank the reviewer for the suggestion: we have extended our experimental evaluation (see appendix I.5) to include incomplete verification results on MNIST, for a network adversarially trained with the method by Wong and Kolter (2018).
> > Therefore, we have assessed the quality of the bounds from Active Set on networks with three different training methods (plain SGD, (Madry et al., 2018), (Wong and Kolter, 2018)) on two different datasets (CIFAR-10, MNIST). For all three experiments (Figures 1, 6, 10) our solver yields better speed-accuracy trade-offs than ``"Gurobi 1 cut".

---

### Official Review · AnonReviewer1 · 2020-10-27
**Tighter optimisation schem for piecewise linear network verification**

**Rating:** 6
**Confidence:** 2

**Review:**

1) Summary
The manuscript proposes an optimisation scheme to compute bounds on the output of piecewise linear feedforward networks. Improving upon the planet relaxation, the authors provide a tighter relaxation scheme with exponentially man constraints that they tackle by a dual solver using active sets. Some experiments are conducted in order to benchmark the relaxation with others.

2) Strengths
+ The paper is mostly well written.
+ The evaluation seems pretty exhaustive as it comes to competing optimisation schemes.

3) Concerns
- The practical utility for deriving a proper bound for a relevant architecture is still limited.
- It does not seem that the paper makes code and data available to the public.

4) Remarks/Questions
  a) Section 2: The optimization problem in Equation (1) needs to be sharpened. On the one hand, $x^hat_n$ needs to be a scalar quantity in order to have a meaningful objective and on the other hand, the only free variable is $x_0$ as all the other variables depend on it. The first part also applies to Equations (2) and (3). Also, the solution is not necessarily unique in the ReLu case if many neurons are silent.
  b) References: capitalization not correct e.g. "smt"
  c) Please provide a more concise statement what you require from the oracle. Do you require that $min_{x in C} a'*x$ can be computed in $O(1)$?
  d) What restricts the approach to piecewise linear functions?

---

> ### Author Response · Authors · 2020-11-17
> **We thank the reviewer for their feedback and address their comments**
>
> We thank the reviewer for their positive assessment and feedback.
>
> **Neural network verification's state-of-the-art.**
> While we agree that the state-of-the-art in neural network verification in general would fall short of formally verifying networks of hundreds of thousands of neurons, we believe that efficient solvers for tighter relaxations are a step forward in this direction.
> This is confirmed by our gains over state-of-the-art methods on complete verification, and by the increased scalability with respect to primal algorithms (Figure 1).
>
> **Code availability.**
> Our full code and data for reproducing all the experiments in the paper was uploaded with the original submission in the supplementary material. The README.md file contains instructions for reproducing the experiments. We will nevertheless also release all our code in a public repository right away upon publication.
>
> **Reviewer's remarks.**
> a) We adjusted the notation to reflect that $\hat{x}_n$ is a scalar. In the formulation of equation (1), activations after the input layer are optimization variables subject to equality constraints, and thus need to be optimized over. This is a consequence of representing the layers as constraints (rather as a composite function in the objective function). Moreover, for verification, we do not need the solution to be unique.
> b) We have fixed the capitalization in the references.
> c) As in previous work (Bunel et al 2020a, Dvijotham et al. 2018), we require that the minimisation of equation (10) can be performed in closed-form. We have now clarified this in the paper.
> d) The restriction to piecewise-linear functions is inherited from the formulation by Anderson et al., for which we provide a dual solver. However, the dual active set framework can be generalised to any LP formulation that features an efficient separation oracle.

---

### Official Review · AnonReviewer3 · 2020-10-29
**A solid contribution to the neural network verification literature**

**Rating:** 7
**Confidence:** 4

**Review:**

The paper proposes a dual approach for solving the exponentially-sized linear programs that arise from the relaxation of the single ReLUs by Anderson et al. (2020). The algorithm is demonstrably faster than existing methods in experiments resembling those from Bunel et al. (2020). The idea of applying a dual active set method is arguably quite straightforward from an optimization perspective. What makes the contribution in this paper strong is

(1) A good implementation in multiple ways. The performance of active set methods highly depends on the how the sets are maintained (section 3.2). Also, making the algorithm run efficiently in practice requires taking advantage of GPUs and the neural network structure (section 3.3, G). I believe that without doing these well, it is likely that such an approach would be significantly slower than existing methods. I look forward to seeing the authors code released.

(2) Providing the proper context for the work -- The paper builds upon prior contributions from Anderson et al. and other works that study the dual aspect of the problem. The authors do a good job explaining how their work fits in and how it compares to other approaches. This both provides a nice theoretical grounding for this work and is also useful pedagogically.

(3) Detailed experiments on reasonably sized neural networks and providing hyperparameters.

Two things I would like the authors to address:

(1) Experimental results - CPU-only

How does an all CPU version of the algorithm compare to Gurobi? This would be a better apples-to-apples comparison since commercial solvers do not make use GPUs. I would understand if a CPU-only implementation is slower since Gurobi is highly-tuned.

(2) Comparison with Tjandraatmadja et al. (2020)

I would like the authors to contrast their approach more against the the one by Tjandraatmadja et al. The formulations used there are different (and the focus of that paper is different), they do describe a cutting plane approach, which in a sense can be viewed as an approach that incrementally increases the number of dual variables. The authors can either do so here in the response, or if they think it would benefit the paper add it in.

-------------------------
Update after author response:
Thanks to the authors for addressing my questions!

---

> ### Author Response · Authors · 2020-11-17
> **We thank the reviewer for their comments, provide CPU-only results, and expand the comparison with Tjandraatmadja et al. (2020)**
>
> We would like to thank the reviewer for their positive comments and feedback.
>
> **Experimental results - CPU-only.**
> The point raised regarding a CPU only comparison is valid and we thank the reviewer for raising it as it will aid in making the comparison more thorough. We have updated the incomplete verification experiments with results for Active Set on 4 CPU threads (the same employed by Gurobi 1 cut). In spite of Gurobi’s heuristics and tuning, Active Set (without any modification from the GPU version) proves to be very competitive with the baseline and it outperforms it in Figure 1. Nevertheless, as pointed out by the reviewer, our method is specifically designed to take advantage of GPU acceleration and should be run there in order to be at its full efficiency.
>
> **Comparison with Tjandraatmadja et al. (2020).**
> Let us refer to the LP formulation by Tjandraatmadja et al. as TAH+, and the one by Anderson et al. as AHT+.
> TAH+ (and the resulting cutting plane algorithm) is indeed very closely related to AHT+. In fact, as shown in their Appendix A.2, TAH+ results from projecting out the $\boldsymbol{z}$ auxiliary variables from AHT+.
> Therefore, the relationship between TAH+ and AHT+ mirrors the one between the Planet and Big-M relaxations. Our dual derivation and the Active Set algorithm could be adapted to operate on the projected relaxations (Planet + TAH+) rather than the unprojected ones (Big-M + AHT+).
> We have updated the paper with this discussion.
>
> **Code release.**
> We are excited to hear that the reviewer is looking forward to our code release. Our full code for replicating all the experiments in the paper was uploaded with the submission in the supplementary material. We will also release all our code in a public repository right away upon publication.

---

### Official Review · AnonReviewer2 · 2020-11-01
**Review for "Scaling the Convex Barrier with Active Sets"**

**Rating:** 7
**Confidence:** 2

**Review:**

The authors present a custom solver for verifying properties of neural networks (such as robustness properties). Prior work for neural network verification relies on generating bounds by solving convex relaxations ("convex barrier"). The authors describe a sparse dual solver for a new relaxation which is tighter (but has higher computational complexity). The solver is represented (for the most part) as standard operations built into pytorch, and so it can be easily run on GPUs (they do require a specialized operator to support masked forward/backward passes, and they describe how this is done efficiently for convolutional networks). The solver involves repeatedly solving modified instances of a problem, where only a small active set of dual variables (instead of exponentially many) is considered at each step.

Experimental results are promising in that it outperforms generic solvers in terms of both the bounds achieved and the time taken to do so. This does seem to be a promising approach.

---

> ### Author Response · Authors · 2020-11-17
> **We thank the reviewer for their comments**
>
> We would like to thank the reviewer for their positive comments and summary of our work.

---

### Official Review · AnonReviewer5 · 2020-11-10
**Review for "Scaling the convex barrier with active sets"**

**Rating:** 8
**Confidence:** 5

**Review:**

This paper presents an algorithm for verification of neural networks with ReLU activations. Essentially, it takes the tightened ReLU relaxation of (Anderson et al. 2020), builds its Lagrangian dual, and then applies a column generation scheme to accommodate the explosion of decision variables. The authors present computational results showing that, due to the amenability of the methods to GPU accelaration, they can produce stronger verification bounds than comparable methods working with weaker relaxations in a modest amount more time. Moreover, they show that the method can be successfully embedded in a branch-and-bound-like framework for exact verification.

I like the paper and think it makes a good contribution to the literature. The paper builds heavily on (Anderson 2020), but the algorithm approach is very different and the authors clearly had to do some work (e.g. rederiving the dual to ensure efficient inner problem solves, making convolutions+masking work on the GPU) to make everything work out. The computational results also seem compelling, though I have some potential concerns about the comparison being made.

My main concern is the use of "Gurobi 1 cut" as the baseline for comparison. Given that there is a one-to-one mapping between cuts in the primal formulation (Gurobi 1 cut) and variables in the dual formulation (the new approach), I am curious why the authors did not choose symmetric generation schemes for the two. Would the solve times be significantly lower if only one cut per layer is added (as in ActiveSet), instead of one per neuron? If so, what benefit do multiple iterations of cut generation offer? Is LP incrementalism or warm-starting used, or is the second LP solved from scratch? Even with these changes, I would imagine that the ActiveSet method still runs (much?) faster than the primal approach, but it's quite possible that the bound improvement would shrink.

Minor comments:
* p1: The phrasing "The main bottleneck of the above approach" is ambigious (which approach?). If the approach includes (ii), then the bottleneck will be the enumeration tree, not the convex subproblems.
* p2: The "which is linearly-sized" reads like it applies to the optimal solution of (2), not the formulation (2) itself.
* p18: I think there are some extra primes in the text of Appendix F.2.

---

> ### Author Response · Authors · 2020-11-17
> **We thank the reviewer for their feedback and address their comments on cut generation and LP incrementalism**
>
> We would like to thank the reviewer for their positive assessment and feedback.
>
> **Cut generation scheme.**
> Adding a new $I_k$ mask to $\mathcal{B}_k$ corresponds to adding $n_k$ variables (or, one constraint per neuron). Therefore, the two generation schemes are indeed symmetric on this specification.  We thank the reviewer for the comment, as it will aid in making the text clearer. We have clarified this in the paper.
>
> **LP incrementalism.**
> We use LP incrementalism for our “Gurobi 1 cut” baseline. For complete verification, the second LP is warm-started from the Big-M LP. For incomplete verification, where each image involves the computation of 9 different output upper bounds, we first compute the 9 Big-M bounds one after the other, then do the same for the tightened bounds with one cut per neuron. We clarified this in Appendix I and thank the reviewer for the question.
>
> **Remaining comments.**
> Minor comments: Regarding the bottleneck of branch and bound, we refer to the fact that a convex sub-problem needs to be solved for each node of the enumeration tree.
> We thank the reviewer for this and other writing-related comments (in particular, for spotting typos in the Appendix). We have updated the paper accordingly.

---

### Official Review · AnonReviewer6 · 2020-11-10
**This paper describes a dual solver for neural network verification. Although the authors present promising empirical results, a precise and rigorous analysis and of the active set strategy is lacking.**

**Rating:** 5
**Confidence:** 3

**Review:**

This paper describes a dual solver for neural network verification. In particular, the authors consider a linear relaxation of neural networks with relu activations and develop an active set based method. Numerical comparisons show that the proposed method provides speed-ups in verifying deep networks.

Although the authors present promising empirical results, a precise and rigorous analysis and of the active set strategy is lacking.  It would be great if the authors can clarify several points raised below.


1. Does the active set approach provide any guarantees on the tightness of the solution? It looks like exponentially many optimization variables in eq 6 are initialized at zero, which provides a lower-bound on eq 2. However, it's not clear if the produced lower-bounds are effective.

2. Does the greedy active set extension strategy converge to an optimal solution of the relaxation? Is the algorithm sensitive to the selection criterion and frequency?

3. The authors employ projected gradient with Adam to maximize the dual function d(\alpha,\beta). Does this approach provably converge to the solution of eq 9? It would be nice to describe the performance of other optimizers, e.g., plain SGD.

4. In eq (1a), is \hat x_n a scalar? It would be better to specify that n_n=1.

5. It looks like the integer constraints z\in \{0,1\} are missing in eq 2?

6. Last paragraph on page 2, by an optimal solution of the problem 2 are you referring to the linear relaxation?

---

> ### Author Response · Authors · 2020-11-17
> **We reply to the reviewer's comments, motivate our design choices and provide experiments on sensitivity to selection criterion and frequency**
>
> We thank the reviewer for the valuable feedback.
>
> **Points 1 and 3.**
> Our initialisation procedure ("Big-M" solver), which operates on problem (15), provably converges to the bounds of the Planet/Big-M relaxation (equation (2)) under strong duality. We clarified our explanation in Appendix B1. In practice, Big-M achieves bounds close to optimality for problem (2) in a time which is competitive with existing solvers for the Planet relaxation: see Figure 2 (a). Its effectiveness is further confirmed on complete verification, Figure 3.
> Analogously, Active Set provably converges to the bounds of the primal of the current restricted dual problem.
> Again, the effectiveness of the produced bounds is demonstrated empirically. In Figure 1, Active Set (AS) yields tighter bounds than “Gurobi 1 cut” in a fraction of its runtime and significantly tightens the bounds from Planet-based methods (“Gurobi Planet”, BDD+, Big-M). This is further confirmed by the performance of AS in complete verification (in spite of the additional computational cost with respect to BDD+).
>
> For what concerns the optimization algorithm: any variant of gradient descent can be employed within Active Set (and Big-M). We clarified this in the paper.
> While, differently from Adam [1], vanilla GD would provably converge to the solution of equation (8), the choice of Adam as optimizer  for the experiments was dictated by empirical reasons. In fact, Adam is less sensitive to the step size schedule and displayed faster empirical convergence in our experiments. This is in line with previous dual solvers on looser relaxations (Bunel et al. 2020a, Dvijotham et al. 2018).
> In order to prove effective on complete verification, we seek to maximise short-term bound improvement rather than to systematically reach optimality.
>
> **Point 2.**
> It is possible to converge to the complete $\boldsymbol{\beta}$ (and then, to the optimal solution of the relaxation) with suitable modifications of the selection criterion (e.g., by converging on the restricted problem after each variable addition and recovering the primal optimal solution via the scheme by (Sherali and Choi, 1996)).
> However, in the context of efficient neural network verification, our goal is to rapidly improve upon the bounds from the Planet relaxation, rather than to converge to the optimality of problem (6).
> As explained in section 3.2, this was the motivation behind our selection criterion design.
> Finally, we would like to stress that adding a number of variables larger than quadratic defeats the purpose of the relaxation by Anderson et al.. In fact, the quadratic “extended” formulation would then be preferable (see equation (26) in appendix F.2).
>
> We added a set of experiments on selection criterion and frequency in Appendix I.4. For the selection criterion, we test against random variable selection (by uniformly sampling the $I_k$ masks). It can be seen that random mask selection only marginally improves upon the Planet relaxation bounds, whereas the improvement becomes significant with our selection criterion (section 3.2). Moreover, our selection criterion is rather robust with respect to variable addition frequency, and tends to perform better if variables are added earlier. This makes it particularly suitable for neural network verification.
>
> **Remaining points.**
> 4. Thanks for pointing this out. We have clarified this in the paper.
> 5. As stated in the first paragraph of section 2.1, we consider the relaxed LP, rather than the IP.
> 6. Yes, as per the previous point.
>
>
> [1] On the Convergence of Adam and Beyond, Sashank J. Reddi and Satyen Kale and Sanjiv Kumar, ICLR 2018

---

### Comment · ~Haoze_Wu1 · 2021-02-23
**Consistency between solver results**

Hi,

Could the authors comment on whether the solvers produce consistent results on the reported benchmarks, and whether the counter-examples found by complete solvers have been verified to be real ones? MILP solvers like Gurobi, on which some of the solvers that the paper evaluated are based,  can produce spurious counter-examples or return infeasible mistakenly if parameters like Feasibility Tolerance is too large (e.g., 1e-6), especially on larger models.

Thanks you!

---

> ### Author Response · Authors · 2021-03-23
> **Comments on consistency between solver results**
>
> Hi Haoze,
>
> Thank you for your interest in our work.
> No inconsistency was found in incomplete verification. However, in order to increase the numerical stability of the Gurobi baselines, we round intermediate bound ranges smaller than 1e-4 (see [1]).
> In complete verification, the employed subset of the dataset from Lu & Kumar (2020) is exclusively composed of UNSAT properties (no counter-examples). In fact, in this work, we are primarily interested assessing the impact of tight lower bounds to problem (1).
> All considered algorithms either correctly returned UNSAT, or timed out. However, the implemented Gurobi MILP baseline returns UNSAT in case of infeasibility, hence we are possibly overstating its verification performance on the considered benchmark.
> In any case, we point out that none of the considered complete verifiers, except ERAN, is sound with respect to floating-point arithmetics.
>
> [1] https://github.com/oval-group/scaling-the-convex-barrier/blob/main/plnn/anderson_linear_approximation.py#L690

---

> > ### Comment · ~Haoze_Wu1 · 2021-04-08
> > **Response to the comment on consistency between solver results.**
> >
> > Ah I see. Thanks a lot for the clarification.

---

### Decision · Program_Chairs · 2021-01-07
**Final Decision**

**Decision:**

Accept (Poster)

**Comment:**

Dear authors,

as you have noticed this paper was not easy to review. I have hence invited 2 additional reviewers which I strongly respect and are very knowledgeable. After carefully reading the paper myself, I have to agree with one of the reviewers who said "... it [your paper] makes a good contribution to the literature ....". To be honest, we were working in my group on a very similar approach but did not manage to finish it (and I know how hard it is).

To conclude, when preparing to the final version, please try to go over the reviews, I am sure they can make your paper even stronger :)